# Deep learning-based age estimation from chest X-rays indicates cardiovascular prognosis

Hirotaka Ieki [1,2,3], Kaoru Ito [1✉], Mike Saji [3], Rei Kawakami[4], Yuji Nagatomo [3,5], Kaori Takada[6], Toshiya Kariyasu[6,7], Haruhiko Machida[6,7], Satoshi Koyama[1], Hiroki Yoshida[1,2], Ryo Kurosawa[1], Hiroshi Matsunaga[1,2], Kazuo Miyazawa[1], Kouichi Ozaki[1,8], Yoshihiro Onouchi [1,9], Susumu Katsushika[2], Ryo Matsuoka[2], Hiroki Shinohara[2], Toshihiro Yamaguchi[2,10], Satoshi Kodera[2], Yasutomi Higashikuni [2], Katsuhito Fujiu [2], Hiroshi Akazawa [2], Nobuo Iguchi[3], Mitsuaki Isobe[11], Tsutomu Yoshikawa[3] & Issei Komuro[2✉]

## Abstract

**Background** In recent years, there has been considerable research on the use of artificial intelligence to estimate age and disease status from medical images. However, age estimation from chest X-ray (CXR) images has not been well studied and the clinical significance of estimated age has not been fully determined.

**Methods** To address this, we trained a deep neural network (DNN) model using more than 100,000 CXRs to estimate the patients' age solely from CXRs. We applied our DNN to CXRs of 1562 consecutive hospitalized heart failure patients, and 3586 patients admitted to the intensive care unit with cardiovascular disease.

**Results** The DNN's estimated age (X-ray age) showed a strong significant correlation with chronological age on the hold-out test data and independent test data. Elevated X-ray age is associated with worse clinical outcomes (heart failure readmission and all-cause death) for heart failure. Additionally, elevated X-ray age was associated with a worse prognosis in 3586 patients admitted to the intensive care unit with cardiovascular disease.

**Conclusions** Our results suggest that X-ray age can serve as a useful indicator of cardiovascular abnormalities, which will help clinicians to predict, prevent and manage cardiovascular diseases.

## Plain language summary

Chest X-ray is one of the most widely used medical imaging tests worldwide to diagnose and manage heart and lung diseases. In this study, we developed a computer-based tool to predict patients' age from chest X-rays. The tool precisely estimated patients' age from chest X-rays. Furthermore, in patients with heart failure and those admitted to the intensive care unit for cardiovascular disease, elevated X-ray age estimated by our tool was associated with poor clinical outcomes, including readmission for heart failure or death from any cause. With further testing, our tool may help clinicians to predict outcomes in patients with heart disease based on a simple chest X-ray.

[1] Laboratory for Cardiovascular Genomics and Informatics, RIKEN Center for Integrative Medical Sciences, Yokohama, Japan. [2] Department of Cardiovascular Medicine, Graduate School of Medicine, The University of Tokyo, Tokyo, Japan. [3] Department of Cardiology, Sakakibara Heart Institute, Tokyo, Japan. [4] Department of Computer Science, School of Computing, Tokyo Institute of Technology, Tokyo, Japan. [5] Department of Cardiology, National Defense Medical College, Tokorozawa, Japan. [6] Department of Radiology, Sakakibara Heart Institute, Tokyo, Japan. [7] Department of Radiology, Tokyo Women's Medical University, Medical Center East, Tokyo, Japan. [8] Division for Genomic Medicine, Medical Genome Center, National Center for Geriatrics and Gerontology, Obu, Japan. [9] Department of Public Health, Chiba University Graduate School of Medicine, Chiba, Japan. [10] Center for Epidemiology and Preventive Medicine, The University of Tokyo Hospital, Tokyo, Japan. [11] Sakakibara Heart Institute, Tokyo, Japan. ✉email: kaoru.ito@riken.jp; komuro-tky@umin.ac.jp

Aging is a term used to describe a correlated set of declines in function with advancing chronological age. Perceived age, or the estimated age of a person, is a robust biomarker for aging. In clinical practice, physicians unconsciously compare perceived and chronological age[1]. Previous clinical studies have revealed that patients with older perceived age, i.e., those who look older than their chronological age, have advanced carotid atherosclerosis[2], reduced bone mineral density[3], and increased mortality[4]. However, in these studies, perceived age was estimated from patient facial images by >10 medical professionals and averaged[2–5]; therefore, significant variation in perceived age was likely. There have been no studies that test whether perceived age is a robust predictor for age-related diseases, including cardiovascular disease. In recent years, machine learning-based methods have been developed to estimate the presence of Alzheimer's disease[6] and coronary artery disease[7] from facial images of patients. Although perceived age is a useful biomarker for age-related diseases and aging, due to privacy and ethical issues it is difficult to obtain facial images of patients in routine clinical practice.

Recently, deep learning has revolutionized the field of machine learning. Deep neural networks (DNNs) are computational models based on artificial neural networks, consisting of multiple layers that progressively extract higher-level features from raw input. DNNs have been shown to exceed human performance in computer vision and natural language processing tasks[8]. They have also been applied to the medical field in dermatology, radiology, ophthalmology, and cardiovascular medicine, and have achieved human physician-level performance, for instance, in classifying images of skin cancer[9], pneumonia detection from CXRs[10], diagnosing retinal disease[11], and arrhythmia classification from electrocardiograms (ECGs)[12,13]. Furthermore, some recent studies have suggested the possibility of using DNNs to learn patterns that humans have difficulty in recognizing[14–16], such as age and sex estimation from ECGs[17] and brain age estimation from magnetic resonance imaging (MRI)[18].

The chest X-ray (CXR) is quick and easy; therefore, it is one of the most commonly used screening tests for a variety of diseases[19]. Despite its simplicity and ease of use, the CXR provides considerable information and is pivotal for the diagnosis and monitoring of cardiovascular and pulmonary diseases such as heart failure, aortic dissection, pneumonia, lung cancer, tuberculosis, sarcoidosis, and lung fibrosis[20]. Because aging[21] and sex difference[22] cause changes in CXR radiological findings, several studies have explored estimating a patient's age from CXR and developing artificial intelligence capable of conducting this task[23–28]. However, the estimation accuracy of those models has not been validated with independent external test data[24,26–28]. Although the association of estimated age with disease prognosis has been suggested in the general population of a cancer screening trial cohort[25], it is still unclear whether estimated age can predict prognosis in populations with cardiovascular disease, especially heart failure. It also remains unclear whether age discrepancy, i.e., the deviation between chronological and estimated age, has any prognostic value for heart failure. Therefore, the clinical significance of estimated age derived from CXRs has not been fully characterized.

We hypothesized that the estimated age from CXRs using deep learning (X-ray age) could be an indicator of aging status. In this study, we sought to develop and train DNNs to estimate patients' age solely from frontal-view CXRs without any additional clinical information and evaluated its estimation performance using a robust method on independent datasets. Because CXRs are widely used, we assumed that they could provide great clinical significance if the extent of aging could be estimated from CXRs, and that X-ray age could be used as a substitute for perceived age. We explored the clinical implications of X-ray age and its characteristics by analyzing the difference between X-ray age and chronological age, defined as years since birth, and the relationship between X-ray age and CXR findings. We applied the developed DNN to the CXRs of patients with heart failure (HF) and examined its relationship to the patient's background, clinical parameters and HF outcome. Additionally, we explored the model's clinical usefulness in a different situation: patients admitted to the intensive care unit (ICU) with cardiovascular disease. These examinations show elevated X-ray age is associated with a worse cardiovascular prognosis.

## Methods

**Dataset acquisition.** Three datasets were used in this study (Fig. 1 and Supplementary Fig. 1). We used the NIH chest X-ray dataset, which comprises 112,120 png images of frontal-view CXRs from 30,805 unique patients. This dataset also includes metadata containing patient age and sex information with up to 15 labels[29]. We excluded 16 CXR images from patients over 100 years of age, since these images were labeled as over 140 years old, which was considered a labeling error. We randomly split the dataset into three groups (training set: 102,029 images from 28,029 patients (91.0%), validation set: 9426 images from 2,523 patients (8.19%); test set: 613 images from 250 patients (0.81%)). There was no patient overlap between the sets to avoid data leakage during model training, which can lead to overestimation of model performance. We also used the JSRT database, which comprises 247 frontal CXR images from 247 Japanese patients[30]. We removed two images for which age information was not available. The JSRT database was used as an independent test dataset to check the generalizability of our model and to determine whether our model can be applied to other populations with different physiques. Data of patients with HF were obtained from our prospective heart failure registry, which enrolled 1562 consecutive patients with acute decompensated HF who were admitted to Sakakibara Heart Institute (Fuchu, Tokyo), a hospital specializing in cardiovascular disease, between November 2011 and December 2017. The diagnosis of heart failure was based on the Framingham criteria[31]. Patients with acute coronary syndrome and isolated right-sided HF were excluded from the study. Conventional clinical variables including age, sex, etiology of HF, risk factors, blood pressure, heart rate, laboratory data, and echocardiographic findings were obtained from the electronic medical records of the study participants. Events of heart failure, re-hospitalization, and death were recorded. Frontal CXRs within 2 days of hospital admission were used in the analysis. Written informed consent was obtained from all the participants before the study. The study protocol was approved by the Institutional Review Board of the Sakakibara Heart Institute (No. 19-092). Patients with cardiovascular disease who were admitted to the cardiovascular care unit were obtained from the MIMIC-IV 1.0 database (https://physionet.org/content/mimiciv/1.0/)[32] and MIMIC-CXR-JPG 2.0.0 (https://physionet.org/content/mimic-cxr-jpg/2.0.0/)[33]. The MIMIC-IV database is a publicly available database comprising health-related data from patients who were admitted to critical care units of the Beth Israel Deaconess Medical Center (BIDMC). MIMIC-IV contains data from 2008 to 2019. Data for patients admitted to the BIDMC intensive care units were extracted from the respective hospital databases. MIMIC-CXR-JPG contains the CXR study information of patients in the BIDMC between 2011 and 2016[33]. The code that generates the descriptive statistics is publicly available (https://github.com/MIT-LCP/mimic-iv and https://github.com/alistairewj/mimic-iv-aline-study). From the data of 76,540 critical care unit admissions, we extracted data of 3,586 patients fulfilling the following criteria: first critical care unit admission, patient age less than 90 years, frontal CXR available in the MIMIC-CXR-JPG database, admission to a cardiovascular care unit (Service in MED, CMED, CSURG and VSURG), and patients

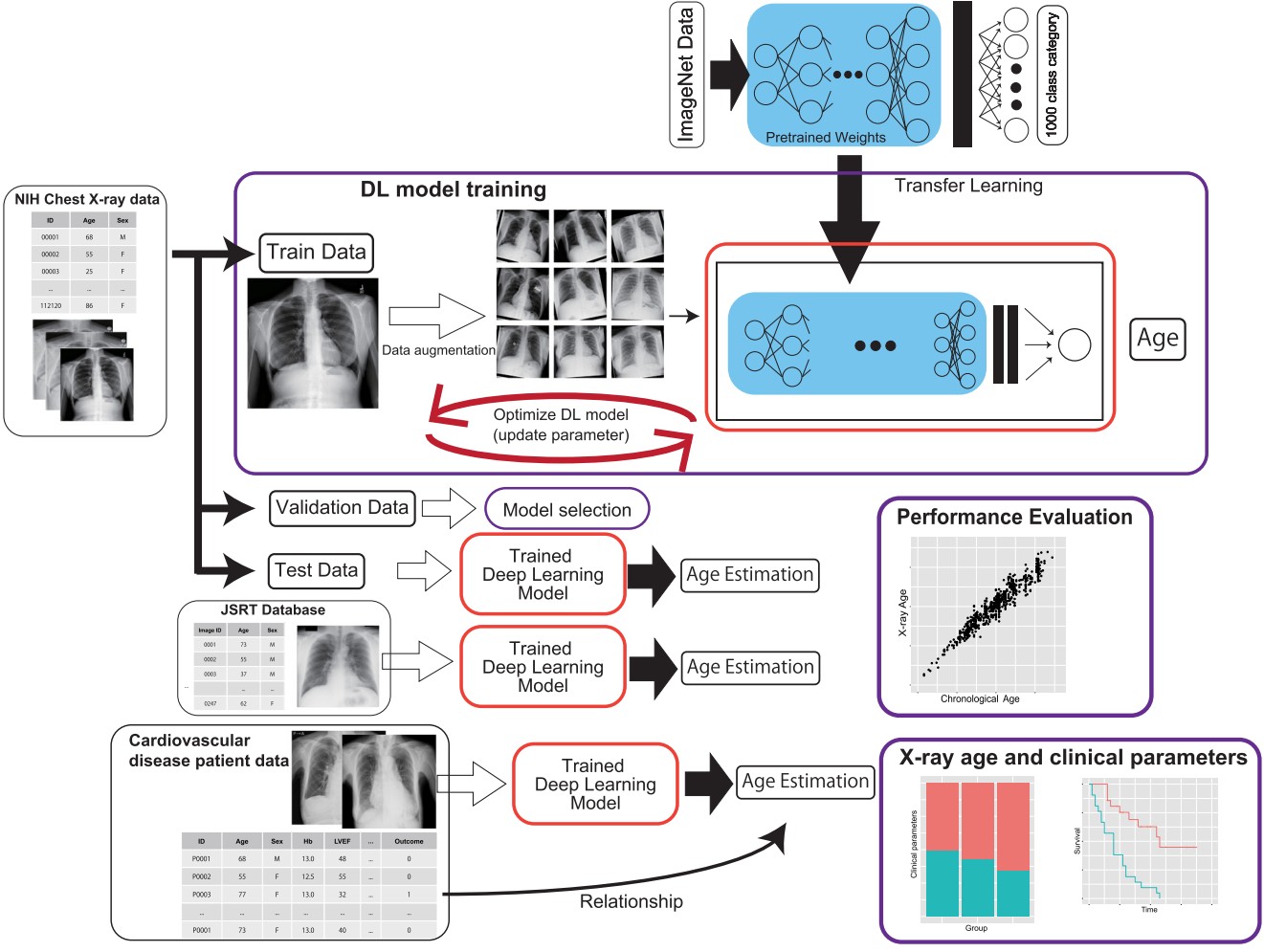

**Fig. 1 Data usage and overall study framework.** The NIH Chest X-ray dataset was randomly divided into training, validation and test datasets. Our deep neural network (DNN) models were trained to estimate the age using the training dataset. The weights of the models were initialized with pre-trained weights on ImageNet data and trained using transfer learning and fine-tuning techniques. Various models with different architectures were separately trained. Validation data were only used to tune the hyperparameters and to select the final model. The accuracy of the deep learning model was estimated using a hold-out test dataset. The independent JSRT dataset was also used to estimate the performance to verify the generalizability of the trained DNN in an independent population. The trained DNN was applied to CXRs of heart failure patients to evaluate the association between the estimated age (X-ray age) and various clinical parameters and the clinical outcomes of heart failure.

with cardiac disease (ICD9 of 410.X, 412.X, 428.X, 425.4-425.9, 398.91, 402.01, 402.11, 402.91, 404.01, 404.03, 404.11, 404.13, 403.91 and 404.93, ICD10 of I21.X, I22.X, I252, I43.X, I50.X, I099, I110, I130, I132, I255, I420, I425, I426, I427, I428, I429, P290).

**Deep learning model development and training**. To develop a deep learning model for age estimation, we applied transfer learning and fine-tuning techniques to our model. We adopted 11 CNN architectures, namely ResNet18, ResNet34, ResNet50, ResNet101, ResNet152[34], DenseNet121, DenseNet161, Dense-Net169, DenseNet201[35], Inception-v4[36], and SENet154[37]. For transfer learning, we used pre-trained weights for CNN models. Pre-trained weights on ImageNet were downloaded for each model from https://github.com/Cadene/pretrained-models.pytorch. Models can be separated into two parts in a CNN: the convolutional and fully connected layer (FCL). Because these models are for the classification task of 1000 categories, the default output layer is comprised of 1000 neurons, which represent the probabilities of each category (Fig. 1). The convolutional layers were initialized with loaded pre-trained weights and frozen. We modified the original FCL part into a new two-layered FCL.

The FCL part is composed of batch normalization, an FCL of 512 neurons with a rectified linear unit (ReLU) as the activation function, batch normalization[38] and a final FCL. Dropout[39] was applied after batch normalization. We adopted an FCL with a single final neuron so that the model outputs a single numerical value of the predicted age and makes it a regression problem. We selected (1) MSE loss, which is defined by the following equations, where $n$ is the number of images, $y_i$ is the actual label, and $\hat{y}_i$ is the estimated age.

$$MSE = \frac{1}{n}\sum_{i=1}^{n}(\hat{y}_i - y_i)^2 \qquad (1)$$

The models were trained on the training dataset to minimize the loss functions. The models were trained using the Adam optimizer and cyclic learning rate policy[40]. During transfer learning, only the parameters in the FCL and batch norm layers of the convolutional part are updated. Then, we fine-tuned the entire network by unfreezing and updating the pre-trained weights with a much lower learning rate. The validation set was used to select hyperparameters to determine when to stop training to avoid overfitting and to select the final model. Validation data were not used to update the weights of the DNN model. The NIH chest X-ray database provides

png images and the JSRT and HF patients' CXRs were DICOM images. All the images were transformed into png images using Python's pydicom library and resized to $320 \times 320$ pixels. To improve the generalizability of our model and avoid overfitting, we applied image augmentation[41]. The images in the training datasets were augmented with random padding and random rotation up to $\pm 20°$. Image flipping was not performed. Our DNNs were trained on NVIDIA Tesla V100 GPUs using a mixed precision training technique[42]. After the training, we selected the model with the lowest loss value in the validation dataset as the final model (Supplementary Table 1). We applied the trained DNN to the test dataset and the JSRT dataset to assess the estimation performance. Inference with the trained model was performed on NVIDIA RTX1080Ti GPUs. Image augmentation was not applied to the test or JSRT datasets. We used gradient-weighted class activation mapping (Grad-CAM)[43] and guided backpropagation[44] methods to visualize the area of interest of our models.

**Age estimation by human physicians**. To compare our model with human physicians' and radiologists' prediction performance, four trained physicians (three cardiologists and one pulmonologist) and three radiologists estimated the patient's age from CXRs on the JSRT dataset. They had 8, 9, 15, 30+, 12, 24, and 26 years of clinical experience, respectively. They estimated the patient's age from the CXR image without any additional information. We used the JSRT data because they are physicians in Japan and are accustomed to analyzing Japanese CXRs. They were allowed to see the training dataset images and labels before estimating age in the JSRT dataset. For ensemble prediction, the age estimates of the four physicians were averaged. For instance, the ensemble prediction is 45-years-old when the four physicians estimated a CXR as a 48-year-old, 52-year-old, 55-year-old, and 25-year-old.

**Statistical analysis of test results**. To estimate the predictive performance of the age estimation model, Pearson's correlation coefficient (r) between chronological age and estimated age was calculated. The correlation coefficient would remain high if the estimated ages were correlated but always estimated higher or lower than the chronological age. To fairly assess the DNN estimation performance, intraclass correlation coefficient case 1 (ICC) and the mean absolute error (MAE) between chronological age and estimated age were also calculated. ICC was calculated using psych package in R. To test the model's reproducibility, we extracted patients who had multiple CXRs within one year in the validation and test data. Age was estimated from CXRs using our DNN and the Pearson's r correlation coefficient was calculated. To remove the possibility of overestimation of estimation performance due to the relatively small amount of test data (0.8% of patients), five-fold cross validation was performed. The NIH chest X-ray data were randomly divided into training, validation, and test datasets in a ratio of 7:1:2, with no overlap of patients among the datasets (Supplementary Table 2). A pre-trained SENet154 model and with the same hyperparameters were used for training. The estimation performance on the test dataset and the JSRT dataset in each split were averaged (Supplementary Table 3). Training of the model using only the No finding data was performed using the 59,998 CXR images from 24,706 patients in the training and validation datasets. The same hyperparameters were used to train the model. The trained model was evaluated using the test dataset and JSRT dataset. Estimation performance was compared between the model trained with all the dataset and the model trained with No finding data only. For the ensemble model, estimated age of the 11 different DNN models was averaged and compared with the SENet154-based single model output. The P value was derived using a 20,000 bootstrap replications method. To analyze the association between the X-ray age and finding labels, linear regression was performed using the validation and test data. Only the first CXR was used for analysis for patients with more than one CXR. The three finding labels of edema, infiltration, and consolidation were grouped together as consolidation and hernia was excluded from the analysis because it was labeled in a small number of CXR images (227 images out of 112,104 images). Regression coefficients of finding labels adjusted for chronological age (Formula in R: *lm(X-ray age ~ finding label + chronological age)*) were determined. Linear regression analysis was also performed to analyze the association between X-ray age and clinical measurements (vital signs, laboratory measurements and past clinical history, such as hypertension, dyslipidemia, diabetes mellitus, smoking history) in the heart failure cohort. Continuous variables were rank normal transformed. The Cox proportional hazards model was used for survival analysis. The median follow-up period was 407 days (interquartile range, 122–879 days). An event was defined as the composite endpoint of heart failure re-hospitalization and all-cause mortality. The independent variables in the Cox model were determined by referring to the empirical rules and previous articles. Age, sex, BMI, history of hypertension, diabetes mellitus, dyslipidemia, and smoking; left ventricular ejection fraction (LVEF); NT-pro BNP; Hb; eGFR and X-ray age by the deep learning model were incorporated as independent variables. In the selection of variables for the multivariate analysis, age, sex and LVEF were fixed as independent variables because they are known to be strong predictors of heart failure outcome[45,46]. Independent variables that showed *P* values of less than 0.05 in the univariate analysis were employed in the multivariate analysis. To compare the Cox model and different independent variables, we used IDI, continuous net reclassification improvement (cNRI), median improvement (MI), and AIC. For the MIMIC data, survival analysis was performed using the Cox proportional hazard model. Events were defined as all-cause mortality. Independent variables in the Cox model were age, sex, eGFR, Hb, diagnosis of congestive HF, diagnosis of myocardial infarction, and age discrepancy (the deviation between chronological and X-ray age). To assess whether X-ray age is clinically useful beyond simply identifying pathological features on CXRs, we also included the CXR abnormality information in the Cox model, using a recently developed abnormality classification DNN[47]. The model output is binary values of whether it is a normal (0) or abnormal (1) CXR and its probability. We calculated the abnormality binary value and its probability from the heart failure cohort's CXRs. We compared Cox model results, including the binary value (abnormality) or logit $[logit(p) = \log(\frac{p}{1-p})]$ of probability (abnormal score) with and without X-ray age information. The R version 3.6.3 base function and 'caret', 'psych', 'survival', 'boot', and 'survIDINRI' packages were used for all statistical analyses. A raw two-sided *p* value is provided when the *p* value is greater than $2.2 \times 10^{-323}$; otherwise, it is provided as $p < 2.2 \times 10^{-323}$ because of generic computational limitations.

**Reporting summary**. Further information on research design is available in the Nature Portfolio Reporting Summary linked to this article.

## Results
**Dataset and model training**. An overview of this study is shown in Fig. 1. First, we used the NIH chest X-ray dataset to develop a DNN that estimates the patient's age from CXR[29]. This dataset is a large publicly available image dataset containing 112,120 png images of frontal-view CXRs from 30,805 unique patients. The dataset also includes metadata containing patient age and sex information with finding labels. After removing individuals with age >100 years because they were considered mislabeled, 112,104

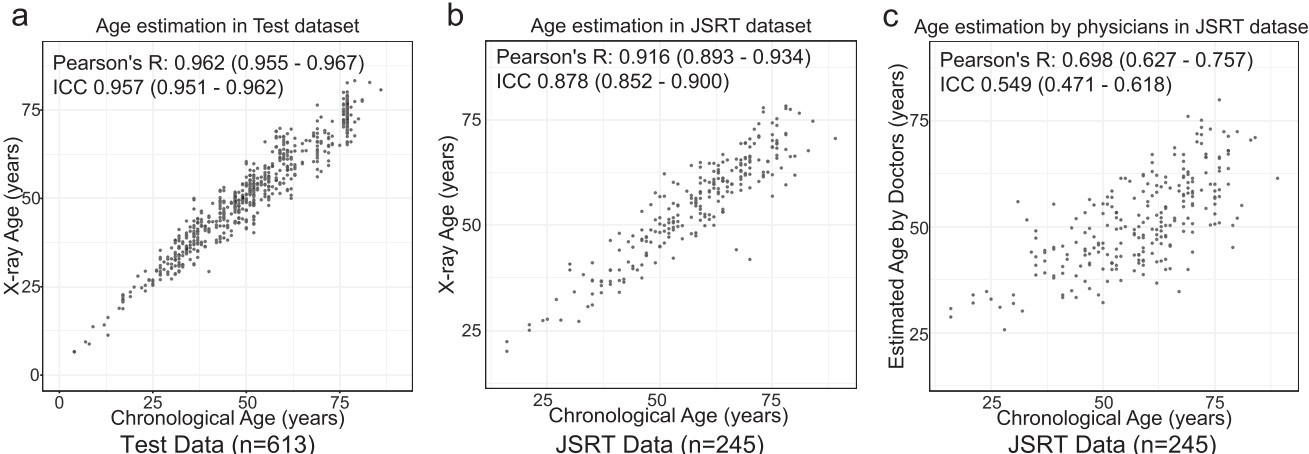

**Fig. 2 Estimation accuracy of the deep learning model and human physicians.** Estimated age by the trained deep learning model (X-ray age) in the test dataset ($n = 613$) (**a**), JSRT dataset ($n = 245$) (**b**), and estimated age by human physicians and radiologists in the JSRT dataset ($n = 245$) (**c**). Scatter plots of the chronological age (x-axis) and estimated age (y-axis) with Pearson's correlation coefficient and intraclass correlation coefficient case 1 (ICC) and 95% confidence interval are shown. A strong positive correlation between the chronological and estimated ages was observed in the deep learning model. For the human estimation, the estimated age is the average of the estimations by the four physicians and three radiologists. The correlation between chronological and estimated ages is modest (**c**).

CXRs remained; of these, 63,328 (56%) were male. The ages ranged between 1 and 95 years, with a median age of 49 years and an interquartile range of 35–59 years (Supplementary Fig. 1a, d). We randomly assigned these data to the training, validation, and test data (Supplementary Fig. 2).

We applied transfer learning and fine-tuning techniques to train the DNN. Briefly, these methods utilize a pre-trained DNN to improve the efficiency of the training time and the amount of data used for training. Rather than training the DNN from scratch, the DNN can learn much faster and with significantly fewer training examples by using transfer learning and fine-tuning[48,49]. We adopted four commonly used architectures, namely ResNet[34], DenseNet[35], Inception-v4[36], and SENet[37] as pre-trained DNNs. To improve the generalizability of our DNN and to avoid overfitting, we applied image augmentation[41]. After the training, we selected the model with the lowest loss value in the validation dataset as the final model. The metrics of the model with the lowest loss (mean squared error [MSE] between the chronological and estimated age) in the validation dataset for each architecture are summarized in Supplementary Table 1. For age estimation, the SENet-based model yielded the lowest loss (MSE loss = 27.34 years[2]) in the validation data (Supplementary Fig. 3). All CXR images in the holdout test dataset were used to measure the performance of the model. The estimated age showed a very strong significant correlation with chronological age (Pearson's r: 0.962 [95% confidence interval (CI), 0.955–0.967]; Intraclass correlation coefficient case 1 (ICC1): 0.957 [95% CI, 0.951–0.962]) and the mean absolute error (MAE) between the estimated age and chronological age was 3.67 (95% CI, 3.44–3.89) years in the test dataset (Fig. 2a, Table 1, Supplementary Data 1).

An important phenomenon known as domain shift sometimes occurs in machine learning, which makes generalization of the machine learning model to unseen data with different distributions difficult[50]. The NIH chest X-ray data were collected from hospitals in the United States[29] and most of the patients were likely to be American. To determine whether our model trained using these data can be applied to other populations with different physiques and from different datasets, we also tested the model on the JSRT dataset, which is a frontal CXR image dataset comprising 247 frontal CXR images from Japanese

patients (Supplementary Fig. 1b, e)[30]. In the JSRT dataset, we also observed a strong significant correlation between the estimated age and chronological age (Pearson's r: 0.916 [95% CI, 0.893–0.934], ICC1: 0.878 [95% CI, 0.852–0.900]), and MAE between the estimated age and chronological age was 4.95 (95% CI, 4.43–5.48) years (Fig. 2b, Table 1, Supplementary Data 2).

To remove the possibility of overestimation of performance due to the relatively small amount of test data, we also conducted a five-fold cross validation (Supplementary Table 2), which showed that the model performance was slightly better when more training data were used (Supplementary Table 3). The performance of the model when trained using only CXRs labeled as No Finding is shown in Supplementary Table 4. The model trained with all CXR images showed better estimation performance compared to the DNN model trained on CXRs labeled as No Finding (Pearson's r: 0.962 vs 0.951, $p < 0.0001$; ICC1: 0.957 vs 0.945, $p = 0.0005$). This is probably due to the fact that in DNN training, the generalization performance of the model improves as the variability of the training data increases[51]. Our DNN estimated patient age more accurately in younger patients than in elderly patients (Supplementary Table 5). We further compared the ensemble inference of eleven different architectures of DNNs with the individual SENet154 model and found no significant difference in prediction performance on either the test or JSRT data (Pearson's r: 0.962 vs 0.960, $p = 1$; ICC: 0.957 vs 0.955, $p = 1$ in the test data; Pearson's r: 0.916 vs 0.912, $p = 1$; ICC: 0.878 vs 0.889, $p = 0.053$ in the JSRT data, Supplementary Table 6). We examined the reproducibility of this model by extracting images taken multiple times for the same patient from the NIH data. The correlation coefficient between the two estimated ages was 0.967 ($p < 2.2 \times 10^{-323}$), indicating that both models also showed high reproducibility (Supplementary Fig. 4). These results suggest that our model can accurately estimate age from CXRs, even in different population groups and cohorts.

**Comparison of predictive performance with human experts.** We compared the predictive performance of our model with that of four experienced physicians and three experienced radiologists using the JSRT dataset. We found a slight correlation between the physicians' estimated age and chronological age, and the average Pearson's correlation coefficient was 0.481 (95% CI, 0.331–0.630).

**Table 1 Summary of estimation performance of DNN in the test and JSRT datasets.**

| Dataset | Test dataset | | | JSRT dataset | | |
|---|---|---|---|---|---|---|
| | Estimate | C.I. | | Estimate | C.I. | |
| | | lower 95% | upper 95% | | lower 95% | upper 95% |
| R | 0.962 | 0.955 | 0.967 | 0.916 | 0.893 | 0.934 |
| MAE | 3.67 | 3.44 | 3.89 | 4.95 | 4.43 | 5.48 |
| ICC | 0.957 | 0.951 | 0.962 | 0.878 | 0.852 | 0.900 |

The age estimation performance of the trained DNN model on the test and JSRT datasets.
R, Pearson's r between the chronological and estimated age, MAE mean absolute error, ICC intraclass correlation coefficient, DNN deep neural network.

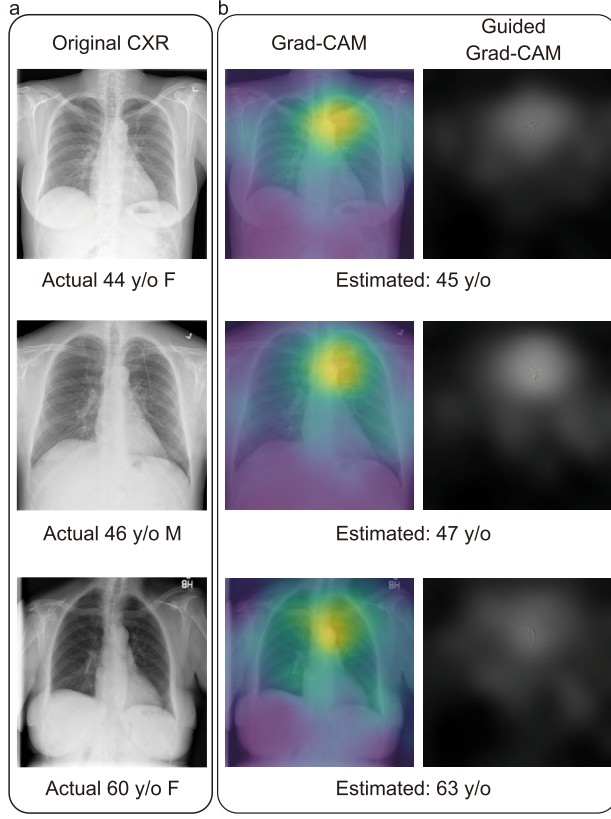

**Fig. 3 Visualization of the deep learning model with Grad-CAM and guided backpropagation.** Example of original CXRs and heatmap visualization using Grad-CAM and guided Grad-CAM. **a** Original CXR image in the dataset with chronological age, sex; F, female; M, male; y/o, years. **b** Visualization of deep learning model using Grad-CAM and a combination of guided backpropagation and the Grad-CAM.

Age estimation performance was better for radiologists than for physicians (Supplementary Table 7, Supplementary Fig. 5). The ensemble predictions by the physicians and radiologists improved the predictive performance (Pearson's r 0.698 [95% CI, 0.627–0.757], MAE 10.06 [95% CI, 9.17–10.94] years); however, this did not match the performance of our DNN (Fig. 2c, Supplementary Table 7, Supplementary Fig. 5, Supplementary Data 3). These results demonstrate that our DNN can learn patterns that are difficult for human experts to recognize.

**Interpretation of the deep learning model by heatmap analysis**. We attempted to visualize the DNN to understand which part of the image it focused on when estimating the patients' age. For this purpose, we created a heatmap using Grad-CAM[43] and guided backpropagation[44]. The model mainly focused on the top of the mediastinum as well as bony features in the sternum, clavicles, and shoulders regardless of the pathology present in the CXRs (Fig. 3 and Supplementary Fig. 6). This is hypothesis provoking that perhaps there are features in these areas (e.g., joint spaces, cartilage or shape and calcification of the aorta) that are important for predicting X-ray age. This pattern is similar to the previously reported CXR-based age prediction model using DNN[24,25], and is consistent with the previous report that tortuosity and calcification of the aorta are hallmarks of atherosclerotic disease and are associated with aging[52–54]. The heatmap analysis results suggest that our DNN models successfully capture changes due to aging.

**A difference between the estimated and chronological age indicates the existence of a disease**. We analyzed CXR images in which the difference between the estimated age and chronological age was large. Some examples of incorrectly estimated CXRs are shown in Fig. 4b. CXRs with a large deviation of estimated age from chronological age seemed to have abnormal findings. When the performance was evaluated using only the CXR labeled No finding, the Pearson's r improved slightly from 0.961 to 0.965 ($p = 0.215$) and the MAE improved from 3.79 to 3.66 years ($p = 0.181$), but was not statistically significant. A significant difference was observed between the estimated and chronological ages when the images had some finding labels. Conversely, we found that CXRs with a substantial difference between the estimated and chronological ages were significantly more likely to have some finding labels, and this tendency increased with age (Fig. 4c, Supplementary Data 4). Regarding each finding label, CXRs with findings of lung fibrosis and effusion were estimated to be significantly older (fibrosis: +1.41 [0.17–2.66] years; effusion: +0.81 [0.10–1.52] years) than the chronological age (Fig. 4d, Supplementary Data 5). These results suggest that a difference between the estimated and chronological age could be a marker for CXR findings, indicating the existence of a disease.

**Estimated age from CXRs (X-ray age) indicates the presence of cardiovascular abnormalities**. To further explore the clinical significance of X-ray age in actual clinical data, we used a private database of patients with acute heart failure (HF). This prospective HF registry has enrolled all patients hospitalized for HF at the Sakakibara Heart Institute since 2011. The registry was designed to collect the clinical background and outcome data of consecutive patients admitted to the Sakakibara Heart Institute for acute decompensated HF. Conventional clinical parameters including age, sex, etiology of HF, risk factors for cardiovascular disease, blood pressure, heart rate, laboratory data, and echocardiographic findings were collected from all study participants ($n = 1562$). The events of HF re-hospitalization and death were also recorded[55–57]. The data comprised 920 (59%) male patients in the age range of 18–98 years, with a median age of 78 years (interquartile range 69–84) (Supplementary Fig. 1c, f, Supplementary Table 8).

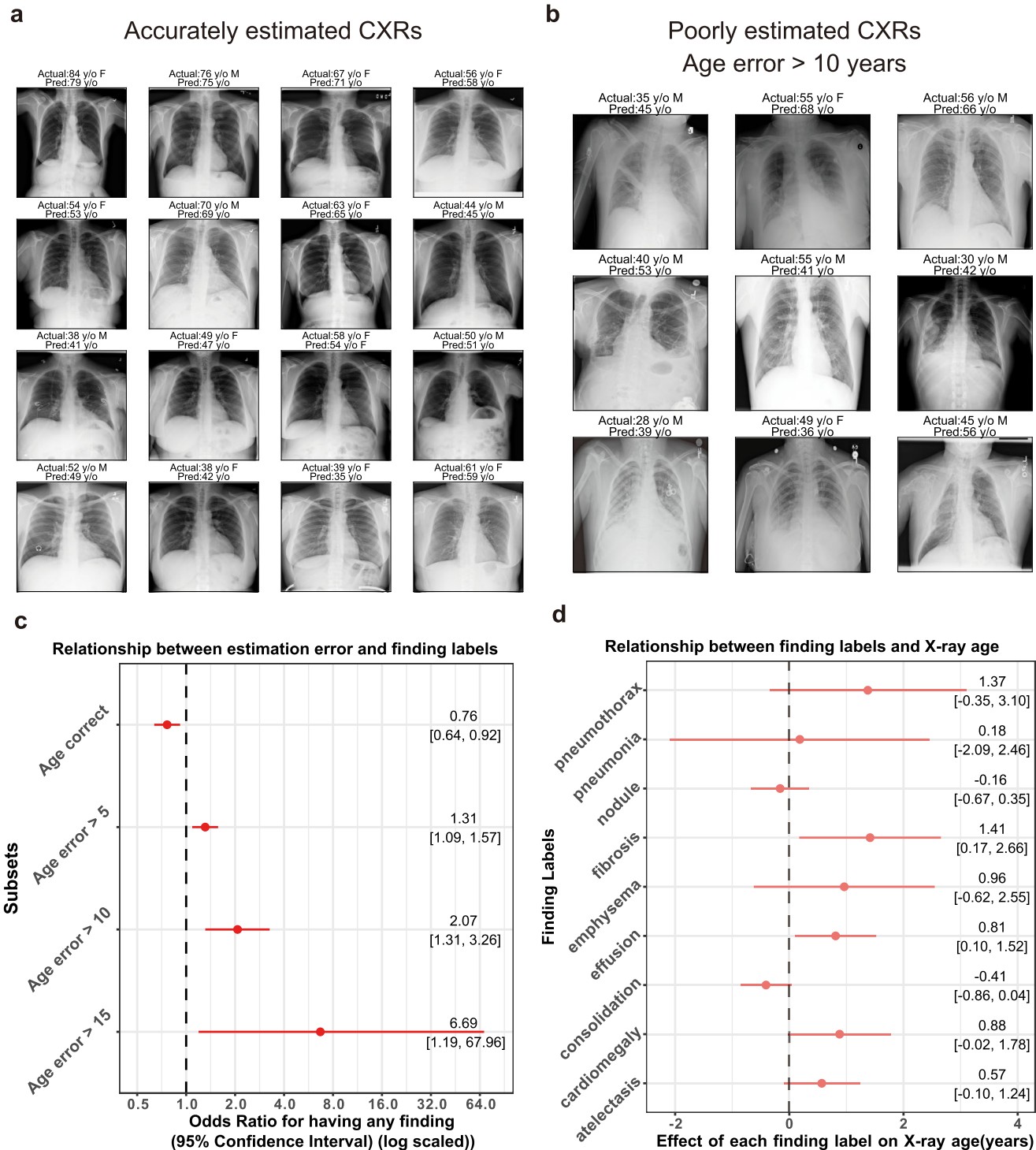

**Fig. 4 Characteristics of images resulting in inaccurate age estimation by the deep learning model. a** Examples of CXR images with an age estimation error of less than 5 years. The chronological age, sex, and X-ray age are shown above each image. Pred, prediction; F, female; M, male; y/o, years. **b** Examples of CXR images with an age estimation error of more than 10 years. **c** Relationship between age estimation error and presence of any finding labels. The odds ratio with a 95% confidence interval is shown on the x-axis ($n = 2773$ independent X-ray images). The odds ratio of having any finding labels was lower in CXR images in which the deep learning model correctly estimated the age. On the other hand, images for which age could not be accurately estimated were significantly more likely to have finding labels. **d** Different finding labels that affect the patient's estimated age. The effect of each finding label on the predicted age derived from linear regression adjusted for chronological age (see Methods) is shown on the x-axis ($n = 2773$ independent X-ray images). For example, CXRs with the 'fibrosis' label are likely to be estimated 1.4 years older than the chronological age.

We applied our model to the CXRs of these patients to estimate their age. Although the performance of age estimation was expected to decrease because all the CXRs were of HF patients and accordingly had some abnormal findings, there was still a significant positive correlation between estimated and chronological age (Pearson's r: 0.769 [95% CI, 0.747–0.789], $p = 4.6 \times 10^{-291}$; ICC1 0.257 [95% CI, 0.218–0.296], $p = 2.1 \times 10^{-25}$). This result suggests that our DNN is still able to estimate aging in the patients with a disease, although its accuracy is reduced. We first examined the association between the patient's history and estimated age from CXR (X-ray age) and found that hypertension and atrial fibrillation were significantly associated with increased X-ray age after adjustment for chronological age (Fig. 5a, Supplementary Data 6). Regarding clinical parameters, increased left atrial diameter on echocardiography, tachycardia and elevated diastolic blood pressure were associated with increased X-ray age, whereas increased weight and taller stature were associated with decreased X-ray age (Fig. 5b, Supplementary Data 7). These significant associations suggest that X-ray age can be an indicator of cardiovascular abnormalities.

**X-ray age predicts cardiovascular prognosis**. Next, we examined the association between HF outcomes and X-ray age. We defined the primary endpoint as the composite endpoint of all-cause mortality and HF re-hospitalization. In the univariate Cox proportional hazards model, X-ray age was associated with the primary endpoint (hazard ratio [HR], 1.040 [per year] [95% CI, 1.031–1.050], $p = 9.29 \times 10^{-17}$) as well as other conventional risk factors such as age, sex, body mass index (BMI), hemoglobin (Hb), NT-pro BNP, and eGFR (Supplementary Table 9). For multivariate analysis, the difference between X-ray age and chronological age was independently associated with the primary endpoint after adjustment for conventional risk factors (HR: 1.019 per 1-year increase of X-ray age [95% CI, 1.005–1.032], $P = 6.69 \times 10^{-3}$), suggesting that patients estimated to be older had a worse HF prognosis (Fig. 5c, Table 2). Compared to the Cox proportional hazard model with conventional risk factors for HF, the model with the addition of age discrepancy as an independent variable significantly increased the predictive performance (continuous net reclassification improvement [cNRI], 0.134 [95% CI, 0.025–0.201], $P = 0.01$; integrated discrimination improvement (IDI): 0.01 [95% CI, 0.0007–0.0245], $P = 0.02$, Supplementary Table 10, Fig. 5d). The Akaike information criterion (AIC) is often used for better model selection and lower values suggest a better model for this criterion. AIC also decreased in the Cox model by adding X-ray age information to the conventional model (7608.6 (nominal model) vs 7206.6, Supplementary Table 10), indicating that X-ray age is an independent prognostic indicator for HF outcome. This additional value of X-ray age in the Cox model was retained, even after adding the CXR abnormality information (see "Methods") to the model (Supplementary Table 10).

We further validated the clinical significance of age discrepancy in patients with cardiovascular disease who were admitted to the ICU at the Beth Israel Deaconess Medical Center in Boston. We extracted 3586 patients with cardiovascular diseases whose CXRs during their ICU stay were available from the MIMIC-IV database (Medical Information Mart for Intensive Care)[32]. The median age of this cohort was 71 years with an interquartile range of 61–80 and 2097 (8.5%) patients were male (Supplementary Fig. 1d, h). Baseline characteristics of this cohort are shown in Supplementary Table 11. In the multivariate Cox model, the difference between X-ray age and chronological age was also significantly associated with all-cause mortality (HR 1017 per 1-year increase in X-ray age [95% CI,

1.0027–1.0305], $P = 1.9 \times 10^{-2}$, Supplementary Fig. 7, Supplementary Table 12).

Finally, we compared the prognostic performance of age discrepancy between younger and older patients. The results showed that the effect of age discrepancy on prognosis was significant in the elderly over 65 years of age (HR 1.024 [95% CI, 1.01–1.038], $p = 9.4 \times 10^{-4}$ (≥65 y.o.), Supplementary Fig. 8). Since the speed of aging varies from patient to patient and the aging effects accumulate with age, this result suggests that the difference in the degree of aging is not so pronounced when patients are young and becomes greater as they get older.

## Discussion

In this study, we verified the performance of our DNN in estimating patients' age from CXRs without any additional clinical data. We also explored the clinical implications of the estimated age. To summarize the main findings of this study: (1) The patient's age was estimated from CXR within 5 years of MAE using a deep learning algorithm. (2) Our DNN estimations of age were much more accurate than the ensemble estimations made by experienced physicians and radiologists. (3) In the heatmap analysis, our DNN successfully captured aging-related changes in CXRs. (4) In the HF population, patients with hypertension and atrial fibrillation were estimated to be older. X-ray age was independently associated with HF outcomes after adjusting for covariates and an association was also observed with prognosis in patients admitted to the ICU for cardiovascular disease. From these findings we conclude that age can be estimated from CXRs with high accuracy and reproducibility using our DNN and that X-ray age can be used as a simple measure to suggest abnormalities and clinical outcome in patients with cardiovascular disease.

Many applications of DNN to automated diagnosis have been studied and human physician-level high accuracy has been reported for various medical images such as skin images, pathology slides, ECGs, CXRs, CT, MRI, and echocardiography[9,10,13,14,58–61]. Several studies have reported that DNN can accomplish tasks that are even difficult for human physicians[14,16,17,62]. In the example of CXRs, Lu et al. created a deep learning model to predict mortality risk from CXR images and stratified the risk of long-term mortality[63]. Toba et al. estimated the pulmonary to systemic flow ratio, an indicator of the severity of congenital heart disease, from CXRs[64]. Since our DNNs age estimation was much better than that of the radiologists, our results also document DNN-learned patterns that were difficult for human experts to recognize.

There have been several studies regarding the estimation of patient age from medical images. A deep learning model that can estimate the age of young adults from MRIs of hands, clavicles, teeth, and knees with high accuracy has been reported[65–68]. Attia et al. created a deep-learning model to predict age and sex from a 12-lead ECG and achieved an MAE of 6.9 years for age estimation. They also reported that patients with a predicted age exceeding the chronological age of more than 7 years had a higher incidence of cardiovascular diseases[17]. Wang et al. proposed a deep learning model to predict patients' age using brain MRI and reported that the estimated age is associated with the future development of dementia[18].

Although aging is associated with CXR findings, few studies have reported age estimation from CXR images[23–28]. Karargyris et al. reported the first convolutional neural network (CNN) model that predicts age from CXR using the NIH dataset. However, they only reported the predictive performance on internal validation datasets, which can lead to overestimation because validation data were used for tuning the hyperparameters of the model. To demonstrate the robustness of the model, its

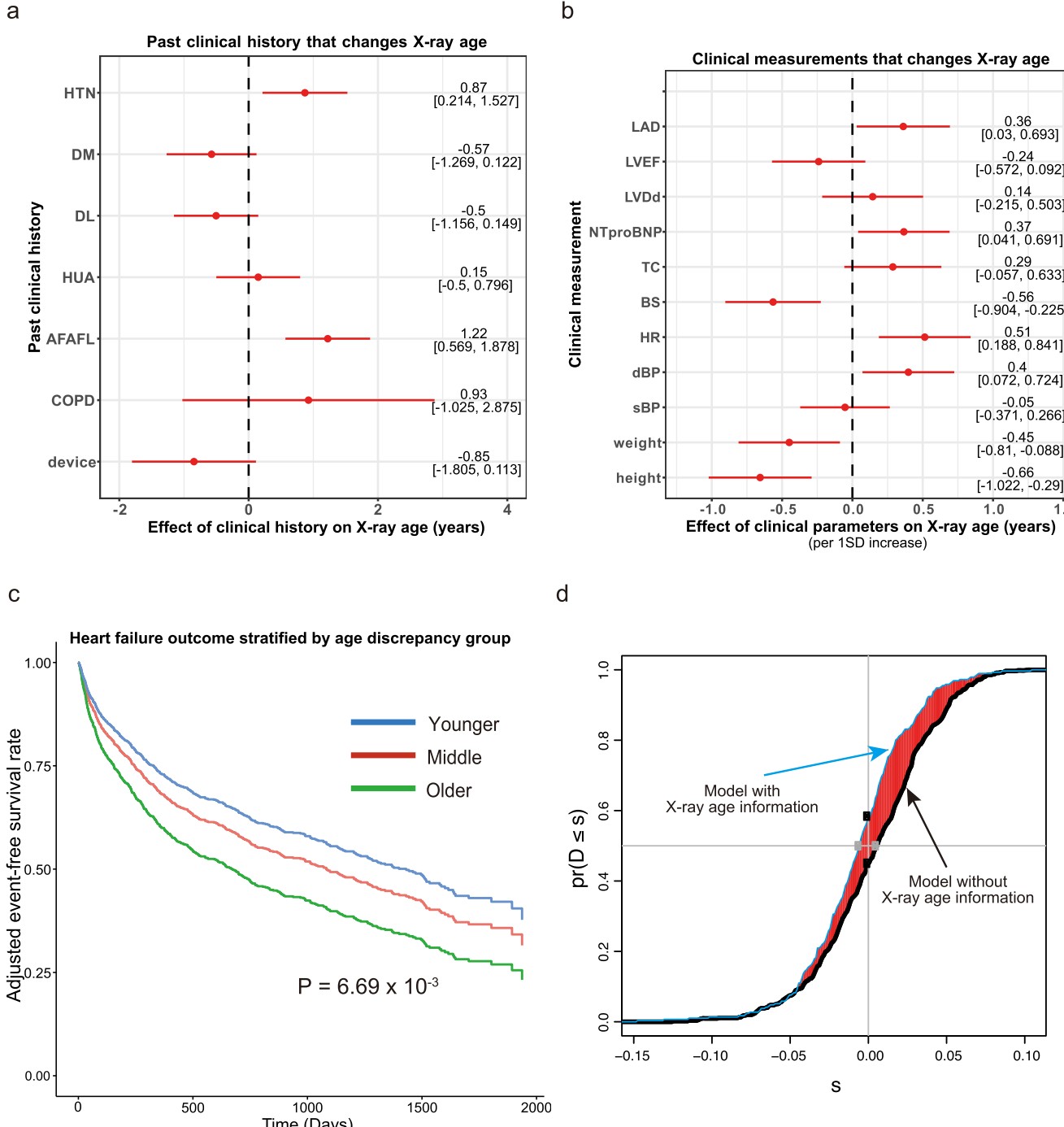

**Fig. 5 Relationship between X-ray age and clinical characteristics and outcome in heart failure patients.** *n* = 1562 study participants. Past clinical history (**a**) and continuous clinical measurements (**b**) affected X-ray age. The effect of specific clinical history or clinical measurements on the predicted age is shown on the x-axis with a 95% confidence interval. For example, the X-ray age of patients with atrial fibrillation or atrial flutter is likely to be estimated 1.22 years older than their chronological age. HTN hypertension, DM diabetes mellitus, DL dyslipidemia, HUA hyperuricemia, AFAFL atrial fibrillation or atrial flutter, COPD chronic obstructive pulmonary disease, device cardiac pacemaker, implantable cardioverter defibrillator, or cardiac resynchronization therapy devices, LAD left atrial diameter, LVEF left ventricular ejection fraction, LVDd left ventricular end-diastolic diameter, TC total cholesterol, BS blood sugar (glucose), HR heart rate, dBP diastolic blood pressure, sBP systolic blood pressure. **c** Adjusted event-free survival curve for heart failure patients stratified by the age discrepancy between the chronological age and X-ray age. An event was defined as the composite endpoint of heart failure re-hospitalization, heart transplantation and all-cause mortality. The top 20% of patients, middle 60%, and bottom 20% were grouped as older, middle, and younger, respectively. **d** Additional value of age discrepancy as assessed by the paired difference of risk scores derived from the Cox proportional hazard model. The figure shows the empirical distribution function of the change in estimated risk score for heart failure patients in the model of conventional risk factors (age, sex, body mass index, LVEF, NT-proBNP, hemoglobin, and estimated glomerular filtration rate) without age discrepancy (thick solid line) and the model with age discrepancy (thin blue solid line). The difference between the areas under the two curves is IDI and the distances between two black dots and between two gray dots are cNRI and median improvement, respectively.

**Table 2 Multivariate Cox proportional hazards model for the primary endpoint.**

| Variable (unit) | Coefficient | HR | Confidence interval | | Z score | P value |
|---|---|---|---|---|---|---|
| | | | lower 95% | upper 95% | | |
| Age (years) | 0.0407 | 1.042 | 1.030 | 1.054 | 6.910 | $4.85 \times 10^{-12}$ |
| Sex (male) | 0.0769 | 1.080 | 0.899 | 1.297 | 0.823 | $4.11 \times 10^{-1}$ |
| BMI (kg/m$^2$) | −0.0272 | 0.973 | 0.950 | 0.997 | −2.239 | $2.51 \times 10^{-2}$ |
| LVEF (%) | −0.0153 | 0.985 | 0.978 | 0.992 | −4.447 | $8.73 \times 10^{-6}$ |
| Log$_{10}$(NT-proBNP) (pg/ml) | −0.1177 | 0.889 | 0.718 | 1.101 | −1.080 | $2.80 \times 10^{-1}$ |
| Hb (g/dl) | −0.1057 | 0.900 | 0.860 | 0.941 | −4.612 | $3.99 \times 10^{-6}$ |
| eGFR (ml·min$^{-1}$·1.73 m$^{-2}$) | −0.0131 | 0.987 | 0.982 | 0.992 | −5.025 | $5.02 \times 10^{-7}$ |
| Age discrepancy (years) | 0.0184 | 1.019 | 1.005 | 1.032 | 2.712 | $6.69 \times 10^{-3}$ |

Coefficients of the Cox proportional hazards model for the primary endpoint in patients with HF.
*HR* hazard ratio, *BMI* body mass index, *LVEF* left ventricular ejection fraction, *Hb* hemoglobin, *eGFR* estimated glomerular filtration rate, *age discrepancy*, difference between the X-ray age and chronological age (X-ray age − chronological age).

performance should be evaluated using unseen data, i.e., an independent dataset, to avoid overfitting and domain shift problems[69,70]. We evaluated the estimation performance on an external test dataset as well as an entirely independent JSRT dataset, neither of which were used during the training phase. Raghu et al. also recently developed a deep learning model to estimate a patient's biological age from CXRs and reported that estimated age was associated with mortality[25]. Although their concept seems similar to ours, there are several key differences. In their model development, Raghu et al. defined biological age as the patient's chronological age plus the difference between the expected death age and the actual death age. Compared to their model, our DNN model is more straightforward in that our DNN is directly trained to estimate the patient's chronological age from a CXR. We showed the generalizability of X-ray age by applying it to other populations with different physiques from an independent dataset. We also showed the additional value of X-ray age by comparing the prognostic models of conventional risk factors with or without X-ray age information. We also demonstrated that the prognostic impact of X-ray age was more pronounced in elderly patients. This is consistent with the fact that the speed of aging varies from person to person and the differences are more pronounced among the elderly.

Our DNN model can be used in several ways in clinical practice. It provides a simple biomarker that represents a single quantification of information from the entire CXR image. Any discrepancy between X-ray age and chronological age suggests the presence of abnormal CXR findings. We found that older patients had a significantly higher probability of hypertension and atrial fibrillation, both of which are related to cardiovascular aging[71,72]. In survival analysis, for example, a 10-year increase in X-ray age has a hazard ratio of 1.20, even after adjusting for other clinical parameters, indicating the clinical significance of X-ray age. Our results suggest that X-ray age has the potential to be used as a simple health indicator, which estimates possible diseases affecting the heart and vessels. As an indicator of the degree of aging, perceived age is a robust biomarker that has been linked to age-related diseases and prognosis. However, a combined dataset of patient facial photographs and clinical information would not be available for research purposes due to ethical and privacy concerns, which hinders the clinical application of perceived age. Furthermore, since the estimated age by a single physician is highly variable and not reproducible, it is necessary to average the estimates by multiple health care providers[2–4]. Since a CXR is used in most patients as a screening test, X-ray age has the potential to replace perceived age as an objective biomarker. In clinical practice, lung age, estimated from spirometry forced expiratory volume (FEV)[73], and vascular age, estimated from carotid artery ultrasonography[74], are used as simple health

indicators and these methods help clinicians explain test results to patients. With continued advances in deep learning, as demonstrated in this study of X-ray age, medical images will also be quantified as age. Several studies have been conducted on automated diagnosis of CXRs; however, in practice, even the same CXR finding can be normal or abnormal considering the age of the patient and it has been difficult to discuss such issues quantitatively. Further research on estimating aging from medical images using DNN may make such quantitative discussions possible.

This study had several limitations. First, all CXR images were obtained from patients; hence, they were obtained for some clinical indications. Further studies are needed to determine if this is applicable to other patients in the general population, such as using a large amount of data from medical checkups. Second, as is often the case with large datasets, the NIH chest X-ray dataset contains low-quality images and labels. Finding labels may not necessarily be accurate because the NIH dataset is labeled using natural language processing[29]. Third, the analysis of our model in HF and MIMIC cohorts is a single-center observational study with a modest number of patients and the findings of the study can potentially include some bias due to its retrospective nature. Fourth, we only examined the relationship between the X-ray age, disease, and prognosis in patients who were hospitalized with heart failure. Thus, these relationships need to be validated in more general and prospective cohorts.

In conclusion, we developed DNNs that accurately estimate patients' age from CXRs without any additional information and with high reproducibility. Our results suggest that estimated age (X-ray age) can serve as an indicator for cardiovascular aging and abnormality and can be a key tool to help clinicians predict, prevent and manage cardiovascular diseases in the era of digital medicine.

## Data availability
The data generated and analyzed during this study are available from the corresponding authors upon request. The NIH chest X-ray dataset used in this study is openly available and can be downloaded at https://cloud.google.com/healthcare/docs/resources/public-datasets/nih-chest. The JSRT database used in this study is publicly available and can be downloaded at http://db.jsrt.or.jp/eng.php. Heart failure patients' data is available upon reasonable request. The MIMIC-IV and MIMIC-CXR-JPG databases used in this study are publicly available and can be downloaded at https://physionet.org/content/mimiciv/1.0/ and https://physionet.org/content/mimic-cxr-jpg/2.0.0/, respectively. Source data for the main figures are available in Table 2, Supplementary Table 10, and Supplementary Data 1–7.

## Code availability
Both our code and trained model for estimating X-ray age are on GitHub (https://github.com/pirocv/xray_age) and archived on Zenodo[75].

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

## Acknowledgements

We would like to express our gratitude to the members of the Sakakibara Heart Institute for their support in collecting samples and providing clinical information. We are grateful to the National Institute of Health, the Japanese Society of Radiological Technology and Physionet for making their data publicly available. This study was supported by the RIKEN management grant. H.I. was funded by the Japan Society for the Promotion of Science grant (JP20J11705, JP22J00780) and the Sakakibara Clinical Research Grant for the Visiting Scientist. H.I., K.I., S.K., and K.M. were funded by the RIKEN management grant. H.I., K.I., S.K., and I.K were funded by the Japan Agency for Medical Research under grant numbers JP20km0405209 and JP20ek0109487.

## Author contributions

H.I., K.I., M.S., Y.N., and T.Y. conceived and designed the study. M.S., Y.N., and T.Y. collected and managed the heart failure patient sample. H.I. and K.I. developed the deep learning model and performed the statistical analyses. R. Kawakami provided computer resources and intellectual advice to develop the deep learning model. K.T., T.K., H. Machida., and N.I. contributed to the clinical interpretation of X-ray images and age estimation. Y.N., S. Koyama, H. Matsunaga, H.Y., R. Kurosawa, K.M., K.O., Y.O., S. Katsushika, R.M., H.S., T.Y., S. Kodera, Y.H., K.F., and H.A. contributed to data analysis and interpretation. K.I., M.I., T.Y., and I.K. supervised the study. H.I. and K.I. wrote the manuscript, and many authors have provided valuable edits.

## Competing interests

The authors declare no competing interests.
