## [Peer Review File · Communications Medicine]

Reviewers' comments:

Reviewer #1 (Remarks to the Author):

The authors present an X-ray Age based on deep learning applied to chest x-ray images trained using >100,000 images. The X-ray age was applied to an external validation population of 1,562 consecutive patients hospitalized with heart failure and 3,586 with cardiovascular disease admitted to the ICU. Elevated X-ray age was associated with worse clinical outcomes in both populations, suggesting that the discrepancy between x-ray age and chronologic age serves as an indicator of cardiovascular abnormalities (though the effect sizes in this study are rather small). Overall a nice study that adds to the growing body of literature that deep learning applied to chest x-rays can lead to useful prognostic biomarkers.

Broad comments:

Throughout the manuscript, I would use “chronologic age” instead of “actual age”, and clearly define “chronologic age” as years since birth. These are standard terms in the biological aging literature.

I am not sure how the sex estimation fits into this manuscript. The majority of the text including the results and discussion are focused on age estimation and discussing aging. I would consider removing sex estimation from this manuscript.

Specific Comments:

Introduction

Line 54 “Perceived age was estimated from facial images of a patient by more than 10 medical professionals and averaged, therefore, it is not an objective index that can be used in clinical practice” –

Why does this make it non-objective? Is it because the perceived age measure was too variable among the 10 medical professionals? Please clarify this.

Lines 79-82 The other studies you cited have already shown that chest x-ray age is associated with disease after adjusting for prognostic factors, and these have been validated on independent testing data. Please clarify your added contribution. The novelty in your paper appears to be the specific use case showing that the chest x-ray age is associated with poor prognosis in patient’s with heart failure/CVD instead of in the general population (which is what the other chest x-ray works have shown using cancer screening trial data)

Results

Line 106-107 “After removing the age outliers” – how did you define an “age outlier” and what is the justification for removing them from the model development data? I see that the answer for this is in the Methods – I would include this brief description in the Results as well for clarity.

Line 154-155 “Models trained with the entire CXR image showed better estimation performance” – I think you mean that models trained using the entire dataset showed better estimation performance (all models are using the full image for training, correct?).

Supplementary Table 7 and Line 158 – You show that the model estimates age way more accurately for those under 65 than those over 65. Could you add a third group to this? <18, 18-64,65+>

I am wondering if the model performance is artificially inflated because it is very easy to predict a child’s age when compared to an adult’s age. For the applications that you suggest, the population younger than 40 or so is probably not meaningful.

Line 208-211 This is confusing. When you start the sentence with, “the difference in age prediction, on the other hand...” makes it sound like the analysis is being done on patients over 20 years old (like the previous sentence). But it seems that this analysis is for using only images without any radiologic findings? Would rewrite the sentence to clarify from the beginning.

Line 217-218 Do you have any intuition why consolidation and effusion lead to a younger chest x-ray age than chronologic age? This seems counterintuitive.

Line 242 How many patients had repeat imaging to calculate the ICC? 0.26 seems fairly low, especially compared to your testing data

Line 244 There is a major confounder here that you are examining associations with comorbidities in patients who are hospitalized with heart failure. Please include this as a limitation in the discussion.

Figure 4e – Is this analysis being done on the raw X-ray age or on the difference between x-ray age and chronologic age? I would do the latter so that you aren’t just reporting associations between findings and age (which we know is correlated with the X-ray age).

Figure 5 – Please increase the font size for your y-axis labels. Can you clarify what summary statistic you are showing in Parts (a) and (b) ? There is a typo in the title of Part a (“history”). In part (c), does this use individuals of all ages? I wonder if this effect would be even larger if you stratified by chronologic age groups. Please add a risk table to the Kaplan-Meier plot to show how many individuals are uncensored at certain timepoints.

Supplementary Figure 2 - For the Heart failure dataset it is unclear why these chest x-rays were obtained. Are these admission CXRs? Were they obtained for any particular indication during stay?

For the MIMIC dataset, no cohort characteristics were provided. Do you have demographic/co-morbidity data for this cohort?

Discussion

“Our evaluation method is robust and fair” – I would avoid this language and instead just state that “we evaluated the estimation performance on an external..”

Line 346-347 The Lu method predicts all-cause mortality, not “mortality risk of lung cancer” – please correct

Line 348-349 The Raghu paper uses CXR-14 and other public cohorts for pre-training – I wouldn't state that the training dataset here is larger.

Line 364-365 "X-ray age can be a simple health indicator that reflects the aging state of the heart and vessels" – It's unknown whether the model is using the heart/vessels to make an age estimation. The black-box nature of deep learning makes it difficult to conclude this.

Line 373-376 the phrase "in clinical practice" is repeated twice.

Reviewer #2 (Remarks to the Author):

Overall, this is a well prepared manuscript which utilizes multiple datasets to develop sex and age prediction models based on chest radiographs and correlate prediction discrepancies with cardiovascular disease risk. This is an interesting strategy for health risk assessment and the conclusions are appropriately caveated with a thorough discussion of limitations. Additionally, the supplementary material is thorough and appropriate.

Comments and Concerns:

1. Consider citing additional references on deep learning and estimation of patient age and sex from chest radiographs:

Sabottke CF, Breaux MA, Spieler BM. Estimation of age in unidentified patients via chest radiography using convolutional neural network regression. *Emerg Radiol.* 2020 Oct;27(5):463-468. doi: 10.1007/s10140-020-01782-5. Epub 2020 Apr 28. PMID: 32347410.

Yi PH, Wei J, Kim TK, Shin J, Sair HI, Hui FK, Hager GD, Lin CT. Radiology "forensics": determination of age and sex from chest radiographs using deep learning. *Emerg Radiol.* 2021 Oct;28(5):949-954. doi: 10.1007/s10140-021-01953-y. Epub 2021 Jun 5. PMID: 34089126.

Yang CY, Pan YJ, Chou Y, Yang CJ, Kao CC, Huang KC, Chang JS, Chen HC, Kuo KH. Using Deep Neural Networks for Predicting Age and Sex in Healthy Adult Chest Radiographs. *J Clin Med.* 2021 Sep 27;10(19):4431. doi: 10.3390/jcm10194431. PMID: 34640449; PMCID: PMC8509558.

2. Line 367: "However, it is not easy to take facial photographs of patients in clinical settings due to privacy concerns, which hinders the clinical application of perceived age." This is not necessarily true. Some healthcare institutions will include photographs of patient's faces in the electronic health record for almost all patients. However, such a dataset would not likely be readily released for research purposes. This statement should perhaps be caveated or modified.

3. After reviewing figure 4, I don't think the discussion which begins on line 393 is necessarily correct:

"Fourth, as described above, an older estimated age does not necessarily mean worse CXR findings. For instance, CXRs with findings of consolidation or effusion were estimated to be younger than the actual age. The transparency of the lung field on CXR increases with age. This is caused by a decrease in the thickness of the chest wall in elderly people. Therefore, we speculate that patients with effusion or consolidation findings could potentially be estimated as younger than their actual age because these findings reduce the transparency of the lung field. This fact implies that, as we demonstrated, not estimated age itself but the dissociation between actual age and estimated age may be important for detecting the presence of a disease."

This speculation does not make sense to me because consolidation and effusion are localizable opacities while the increased transparency associated with aging would impact the bilateral lung fields diffusely. The neural networks should not be conflating effusions and consolidations with diffuse changes in penetration associated with differences in chest wall thickness. My speculation is that effusion and consolidation lead to younger predictions based on a bias in the clinical reason for obtaining chest radiographs in young patients. Younger patients in a hospital setting may be less likely to be imaged with a chest radiograph if they do not have cardiopulmonary symptoms. However, pneumonia, consolidation, or effusion may warrant repeated imaging. I think that a supplementary figure or table may be helpful to clarify this issue. Perhaps a figure can be constructed which shows the number or percentage of cases with various findings such as consolidation or effusion broken down based on 5 year intervals of patient age. This would be helpful to better understand the data being presented in figure 4e.

Reviewer #3 (Remarks to the Author):

In this paper, the authors present a deep learning based computer vision system for estimating "chest X-ray age" from frontal chest radiographs and show that this "chest X-ray age" is predictive of clinically meaningful outcomes and that it is additive to traditional risk factors. The authors also present a model to predict sex based on chest radiographs, but this application is far less interesting/clinically applicable and seems to detract from the main thesis of the paper.

There are many strengths of this paper – it is methodologically sound and crucially, it was validated/tested on a number of external datasets demonstrating its generalizability to different patient cohorts which is so important in an era where awareness of the pitfalls of dataset shift is growing. This sets this work apart from prior similar efforts. Additionally, the fact that the authors' model is capable of accurately estimating patient age on chest radiographs, a task that the authors demonstrate eludes human interpreters in their comparison to a consensus of experienced X-ray interpretations, is a fascinating example of the power of deep learning technologies to unlock hidden patterns in complex data.

Major concerns:

Despite the strengths of this work as noted above, I do have concerns about the overall premise of the paper – namely that the discrepancy between predicted age by the deep learning model and actual age is predictive of clinically meaningful outcomes. Each of the CNN architectures explored in this paper were trained using the label of actual chronological age – therefore the difference between predicted age and actual age is essentially a measure of the error of the model. I understand the author’s reasoning that this may be a clinically meaningful metric – in an appropriately fit model that fits to features in the chest radiograph that might suggest an older age, then the model might have a larger error of prediction on those chest radiographs that have more of these features, suggesting a worse clinical prognosis. However, I have significant concerns about this assumption.

First, this premise is highly contingent on the training process itself and the dataset on which the model was trained. As an example, if one were to be able to theoretically train a nearly perfectly performing model (for example using a different architecture) while still not sacrificing generalizability to external datasets, then this discrepancy between predicted age and actual chronological age would disappear. In this case, the better performing model might fit to more robust features in the chest X-ray that are associated with chronological age regardless of the pathology present on chest X-ray. It seems that in this case the authors were fortunate to train a model that seemed to fit to features associated with pathology on the chest radiographs as they demonstrated in their analysis of the association between chest X-ray/chronological age discrepancy and pathological labels in the NIH CXR dataset.

The frailty of this assumption is also demonstrated by the variable performance of the model on the various datasets under study in this paper. For example, the performance of the age prediction model in the HF dataset was much lower than in other datasets and the authors theorize that this may be because there is more pathology present in this dataset and thus greater discrepancy between the model’s predicted age and actual age. However, what would happen if the model was TRAINED on the HF dataset? In minimizing loss/error on this dataset, the model would likely learn to ignore some of this pathology present on chest radiographs and instead focus on other features that might be associated with age.

Expanding further on this point, I am unsure about the clinical utility of a model that predicts “chest X-ray age”, when this model may essentially be fitting to pathological features present on chest radiographs. Would a model that is trained to directly identify pathology on chest radiographs be a stronger predictor of outcomes?

In order to prove that “chest X-ray age” is clinically useful beyond identifying pathological features present on chest radiographs, my recommendation is that the authors train a separate model to identify pathologic labels (e.g. cardiomegaly, pleural effusion, etc.) on the NIH CXR dataset. While evaluation of the accuracy of these identified labels might not be possible on some of the external datasets due to lack of ground truth labels, the authors can still explore whether these automatically generated labels might be predictive of primary composite outcomes (e.g. composite outcome of HF hospitalization or mortality in the HF cohort). The authors could explore the additive predictive value of individual pathological labels and/or ANY pathological label in multivariate modeling. The authors could also explore whether “chest X-ray age” retains significance in a multivariate Cox model adjusted for these labels or that introducing chest X-ray age into a model adjusted for these labels still improves AIC/BIC in order to definitively conclude that “chest X-ray age” is a clinically meaningful metric.

Minor concerns:

1. As above, I am unsure what the sex prediction model adds to this paper. This model does not appear to be clinically meaningful and sex prediction does not appear to be a very interesting task as the authors demonstrate that this model seems to fit to breast shadows on the chest radiograph. My suggestion would be to remove this model completely from the paper as it detracts from the study's main message.
2. Lines 118-119 – Why not use an ensemble of all models together rather than choosing the best performing model? There is evidence that model ensembling can help resolve the bias-variance trade-off and may lead to better generalizability across diverse datasets. You could compare performance of an ensemble of all models (or X top performing models) vs each individual model to explore this possibility.
3. The idea that consolidation and effusion were associated with younger CXR age than chronological age is counterintuitive to me. I appreciate the authors' attempt at explaining this phenomenon in the discussion, however, one would think that there are enough examples of consolidation or effusion in the NIH CXR dataset such that the model could learn to differentiate focal opacities associated with effusion or consolidation vs. generally lower transparency of the lung fields in younger patients. This explanation is therefore not very satisfying for an intriguing finding. Perhaps, the authors could provide a more in depth analysis of visualization techniques such as Grad-CAM and guided backprop on some examples of these chest radiographs in order to explore which features the model might be using to predict a lower "chest-Xray age".
4. In the multivariable Cox models, there appears to be a difference between the nominal models used to evaluate the significance of CXR age and Age_discrepancy (Supplementary table 12). Why was chronological age left out of the baseline model for CXR age? Please provide an additional multivariable model analyzing the significance of CXR age in a multivariable model adjusted for all of the baseline variables including chronological age.
5. Code availability – it seems a shame that the code is available "on request". It doesn't seem that there are proprietary concerns with this model (given the authors used publicly available CNN architectures) and the model was trained on publicly available datasets. For the benefit of the scientific community and to allow external validation of this model across diverse datasets by interested scientists, it is my opinion that both the code to generate this model and the trained weights of the best performing model(s) should be made publicly available on a platform such as Github, Bitbucket, or Sourceforge if the paper is to be accepted for publication.

First, we sincerely thank all the reviewers for investing a considerable amount of time to provide these high-quality comments, which have helped us to considerably strengthen and better our manuscript. We reanalyzed our data throughout, therefore some of the numbers have changed slightly, but the conclusions remain the same. Major changes in the revised manuscript are summarized below.

- 1) The sex inference model that is not the main point of our paper has been removed. Accordingly, our manuscript has been reorganized
- 2) We have re-analyzed and conducted additional analyses to discuss the result that "consolidation" and "effusion" tend to make X-ray age younger, and the impact of such disease labels on X-ray age.
- 3) We added analyses to investigate the effect of chronological age (e.g. younger or older) on the predictive performance.
- 4) We have examined whether our X-ray age, in addition to CXR disease labels, is useful for prognostic stratification of patients.
- 5) The "Code availability" section has been created, which indicates the URL for downloading our code.

Also, please see below for the responses to each reviewer's comment not mentioned above. We thank all the reviewers for these valuable comments and the opportunity to improve our paper.

REVIEWER COMMENTS

Reviewer #1 (Remarks to the Author):

The authors present an X-ray Age based on deep learning applied to chest x-ray images trained using >100,000 images. The X-ray age was applied to an external validation population of 1,562 consecutive patients hospitalized with heart failure and 3,586 with cardiovascular disease admitted to the ICU. Elevated X-ray age was associated with worse clinical outcomes in both populations, suggesting that the discrepancy between x-ray age and chronologic age serves as an indicator of cardiovascular abnormalities (though the effect sizes in this study are rather small). Overall a nice study that adds to the growing body of literature that deep learning applied to chest x-rays can lead to useful prognostic biomarkers.

Response: We appreciate the reviewer's careful assessment of our manuscript and are grateful for the numerous constructive comments and advice the reviewer has provided.

Accordingly, we have revised our manuscript to enhance both its validity and readability. The following is a point-by-point response to each of the reviewer's comments:

Reviewer #1, Broad comments #1

Throughout the manuscript, I would use “chronologic age” instead of “actual age”, and clearly define “chronologic age” as years since birth. These are standard terms in the biological aging literature.

Response: Thank you very much for pointing out our inappropriate use of the term "actual age". Accordingly, we have modified our manuscript to use “chronological age” instead of “actual age” throughout and included the definition in the main text as follows.

Lines 92-94: “...analyzing the difference between X-ray age and **chronological** age (defined as years since birth) and the relationship between X-ray age and CXR findings.”

Reviewer #1, Broad comments #2. I am not sure how the sex estimation fits into this manuscript. The majority of the text including the results and discussion are focused on age estimation and discussing aging. I would consider removing sex estimation from this manuscript.

Response: We appreciate the reviewer's important remarks. As you indicated, since the most crucial part of our study is the age estimation model and the results, we have decided to follow the reviewer's advice and excluded the section on sex estimation from our manuscript. We believe that this change has resulted in a more concise organization of our manuscript and has made it easier for the readers to grasp the main point.

Reviewer #1, Specific comment #1. Introduction:

Line 54 “Perceived age was estimated from facial images of a patient by more than 10 medical professionals and averaged, therefore, it is not an objective index that can be used in clinical practice” –

Why does this make it non-objective? Is it because the perceived age measure was too variable among the 10 medical professionals? Please clarify this.

Response: We apologize for our insufficient explanation. While there can be variation in the perceived age estimated by health care professionals from a facial image, this does

not mean that it is not objective. On the other hand, because it could be difficult to collect judgments from more than 10 medical professionals in actual clinical practice, we have modified those sentences as follows.

Lines 50-53 “However, in these studies, perceived age was estimated from facial images of a patient by more than 10 medical professionals and averaged ²⁻⁵; therefore, variation in perceived age may increase. Perceived age is also difficult to implement in actual clinical practice because it is difficult to obtain decisions by more than 10 medical professionals.”

Reviewer #1, Specific comment #2. Introduction::

Lines 79-82 The other studies you cited have already shown that chest x-ray age is associated with disease after adjusting for prognostic factors, and these have been validated on independent testing data. Please clarify your added contribution. The novelty in your paper appears to be the specific use case showing that the chest x-ray age is associated with poor prognosis in patient’s with heart failure/CVD instead of in the general population (which is what the other chest x-ray works have shown using cancer screening trial data)

Response: We appreciate the reviewer’s constructive comment. Accordingly, we have modified our manuscript to clarify the novelty of our study.

Lines 80-83 “Although an association between estimated age and prognosis has been suggested in the lung cancer patients cohort²⁵, it is still unclear whether there is an association between estimated age and cardiovascular disease and its prognosis, or whether estimated age has any additional value after adjusting for other prognostic factors.”

Reviewer #1, Specific comment #3. Results:

Line 106-107 “After removing the age outliers” – how did you define an “age outlier” and what is the justification for removing them from the model development data? I see that the answer for this is in the Methods – I would include this brief description in the Results as well for clarity.

Response: We thank the reviewer for this suggestion. Accordingly, we included a brief

explanation of removing the “age outlier” in the Results section.

Lines 106-108 “After removing the age outliers (since they are considered mislabeled, see Methods for details), 63,328 (56%) of the 112,104 X-rays were male CXRs.”

Reviewer #1, Specific comment #4. Results:

Line 154-155 “Models trained with the entire CXR image showed better estimation performance” – I think you mean that models trained using the entire dataset showed better estimation performance (all models are using the full image for training, correct?).

Response: We thank the reviewer for this question and apologize for the insufficient explanation. We compared our DNN model with another DNN model that was trained on “No finding” labeled CXRs (about 50% of NIH data). Our model showed better estimation performance compared to the aforementioned “No finding” only model. We have added a sentence to clarify this point.

Lines 148-150 “The model trained with the entire CXR image showed better estimation performance compared to the DNN model trained on CXRs labeled as “No Finding” (Pearson’s r: 0.962 vs 0.951, $p < 0.0001$; ICC1: 0.957 vs 0.945, $p = 0.0005$).”

Reviewer #1, Specific comment #5. Results:

Supplementary Table 7 and Line 158 – You show that the model estimates age way more accurately for those under 65 than those over 65. Could you add a third group to this? <18, 18-64,65+>

I am wondering if the model performance is artificially inflated because it is very easy to predict a child’s age when compared to an adult’s age. For the applications that you suggest, the population younger than 40 or so is probably not meaningful.

Response: Thank you for this constructive suggestion. We tested the model’s estimation performance in the ≤ 18 and 18-64 age groups. As the reviewer expected the estimation performance was better in the ≤ 18 yo group. Thus, we added a “ ≤ 18 y/o” group to the table (Supplementary Table 5). The result indicates that the younger age group is relatively more accurate in predicting age than the older age group. This is expected to be due to the accumulation of variations over time among older individuals.

Next, to address whether our model is meaningless or not for the population under 40 years in terms of our purpose, it is necessary to examine the disease group under 40 years of age. Unfortunately, the heart failure and CCU cohorts used in our study comprised of older individuals with a mean age of 78 (69-84) and 71 (61-80) years, respectively, which did not allow us to tackle the issue. This will need to be investigated in the future using another independent disease cohort with young individuals.

Supplementary Table 5. Age estimation performance in different patient age groups.

Dataset	Subset	R			MAE			ICC		
		Estimate	C.I.		Estimate	C.I.		Estimate	C.I.	
			lower 95%	upper 95%		lower 95%	upper 95%		lower 95%	upper 95%
Test	≥ 65 y/o	0.718	0.624	0.792	4.47	3.91	5.02	0.472	0.353	0.576
	18 y/o <, < 65 y/o	0.921	0.906	0.933	3.45	3.21	3.69	0.915	0.902	0.927

	≤ 18 y/o	0.954	0.870	0.984	3.34	2.57	4.12	0.793	0.572	0.907
JSRT	≥ 65 y/o	0.500	0.322	0.644	6.78	5.64	7.93	0.130	-0.048	0.300
	< 65 y/o	0.904	0.871	0.929	3.96	3.50	4.20	0.893	0.863	0.917

Different metrics of age estimation performance in older (≥ 65 years old), middle-aged ($18 \text{ y/o} <, < 65 \text{ y/o}$), and younger (≤ 18 years old)

patients in the test and JSRT datasets. There were no participants under the age of 18 in the JSRT dataset (Supplementary Fig.1). R,

Pearson's r between the chronological and estimated age; MAE, mean absolute error; ICC, intraclass correlation coefficient.

Reviewer #1, Specific comment #6. Results:

Line 208-211 This is confusing. When you start the sentence with, “the difference in age prediction, on the other hand...” makes it sound like the analysis is being done on patients over 20 years old (like the previous sentence). But it seems that this analysis is for using only images without any radiologic findings? Would rewrite the sentence to clarify from the beginning.

Response: We apologize for this confusing description and we agree with the reviewer’s indication. Accordingly, we have modified the sentence as follows.

Lines 195-198 “When the performance was evaluated using only the CXR labeled “No finding”, the Pearson’s r improved slightly from 0.961 to 0.965 ($p = 0.215$) and the MAE improved from 3.79 to 3.66 years ($p = 0.181$), but was not statistically significant.”

Reviewer #1, Specific comment #7. Results:

Line 217-218 Do you have any intuition why consolidation and effusion lead to a younger chest x-ray age than chronologic age? This seems counterintuitive.

Response:

We thank the reviewer for raising this important point. As the reviewer pointed out, it is counterintuitive that consolidation and effusion lead to a younger X-ray age estimation. We re-analyzed this by removing the duplicated CXRs (multiple CXRs per patient) since duplicated or related CXRs biased the result of a linear regression model with an “independent sample” assumption. As shown in the revised **Fig. 4d** below, “fibrosis” and “effusion” were associated with older X-ray age estimation, while no label was associated with younger X-ray age estimation. We revised our manuscript and the figure below. Please also refer to the response to **Reviewer #2, Major comment #3, and Reviewer #3, Minor comment #3**.

Lines 202-204 “Regarding each finding label, CXRs with findings of lung fibrosis and effusion were estimated to be significantly older (fibrosis: + 1.41 [0.17-2.66] years; effusion: +0.81 [0.10 – 1.52] years) than the chronological age (**Fig. 4d**).”

Lines 513-515 “To analyze the association between the X-ray age and finding labels, linear regression was performed using the validation and test data. Only the first CXR was used for analysis for patients with more than one CXR.”

Fig. 4 Characteristics of images resulting in inaccurate age estimation by the deep learning model.

a, Example of a CXR image with an age estimation error of less than 5 years. The chronological age, sex, and X-ray age shown above each image. Pred, prediction; F, female; M, male; y/o, years. **b**, Example of a CXR image with an age estimation error of more than 10 years. **c**, Relationship between age estimation error and presence of any finding labels. The odds ratio with a 95% confidence interval is shown on the x-axis. The odds ratio of having any finding labels was lower in CXR images, in which the deep learning model correctly estimated their age. On the other hand, images for which age could not be accurately estimated were significantly more likely to have finding labels. **d**, Different finding labels that affect the patient's estimated age. The effect of each finding label on the predicted age derived from linear regression

adjusted for chronological age (see Methods) is shown on the x-axis. For example, CXRs with ‘fibrosis’ are likely to be estimated 1.4 years older than the chronological age.

Reviewer #1, Specific comment #8. Results:

Line 242 How many patients had repeat imaging to calculate the ICC? 0.26 seems fairly low, especially compared to your testing data

Response:

We thank the reviewer for raising this important point and apologize for the insufficient explanation in our manuscript. ICC value was calculated between chronological and estimated age (X-ray age). If X-ray age correlates—but is always estimated lower or higher than chronological age—Pearson’s r value will be still high. Therefore, we think Pearson’s r alone is insufficient for assessing the estimation performance. We thus examined the reproducibility of the X-ray age using data from patients with repeated CXR images (Supplementary Fig. 4). Since the heart failure cohort is a disease cohort and the CXR in the cohort is more likely to have some findings, we speculate that the ICC is lower in this group than that in the test data. Nevertheless, the correlation is still high, indicating that our DNN estimates aging to some extent, although the performance is somewhat reduced. We have clarified this in our manuscript as follows.

Lines 223-228 “Although the performance of age estimation was expected to decrease because all the CXRs were of HF patients and accordingly had some abnormal findings, **there was still a significant positive correlation between estimated and chronological age (Pearson’s r : 0.769 [95% CI, 0.747–0.789], $p = 4.6 \times 10^{-291}$; ICC1 0.257 [95% CI, 0.218–0.296], $p = 2.1 \times 10^{-25}$). This result suggests that our DNN is still able to estimate aging in the patients with a disease, although its accuracy is reduced.**”

Lines 491-495 “**The correlation coefficient would remain high if the estimated ages were correlated but always estimated higher or lower than the chronological age. To fairly assess the DNN estimation performance, intraclass correlation coefficient case 1 (ICC) and the mean absolute error (MAE) between chronological age and estimated age were also calculated.**”

Reviewer #1, Specific comment #9. Results:

Line 244 There is a major confounder here that you are examining associations with co-morbidities in patients who are hospitalized with heart failure. Please include this as a limitation in the discussion.

Response:

We appreciate the reviewer’s important comment, which we agree with. There may be a confounder in the analysis because we examined associations of X-ray age and co-morbidities in a heart failure patient cohort. We have added this to our study limitations as follows.

Lines 378-380 “Fourth, we only examined the relationship between the X-ray age, disease, and prognosis in patients who were hospitalized with heart failure. Thus, these relationships need to be validated in more general and prospective cohorts.”

Reviewer #1, Specific comment #10. Results:

Figure 4e – Is this analysis being done on the raw X-ray age or on the difference between x-ray age and chronological age? I would do the latter so that you aren’t just reporting associations between findings and age (which we know is correlated with the X-ray age).

Response:

Thank you for this important comment. We also apologize for our insufficient explanation of this analysis. The analysis was performed on X-ray age, but adjusted for chronological age. In concrete, the linear regression formula in R language is $lm(X\text{-ray age} \sim \text{effusion} + \text{chronological age})$. Therefore, this analysis does not simply report the relationship between finding labels and X-ray age as per the reviewer’s concern, but the effect of finding labels on X-ray age. We revised our manuscript as follows.

Lines 518-520 “Regression coefficients of finding labels adjusted for chronological age (Formula in R: $lm(X\text{-ray age} \sim \text{finding label} + \text{chronological age})$) were determined.”

Fig.4 Legend “... **d**, Different finding labels that affect the patient’s estimated age. The effect of each finding label on the predicted age derived from linear regression adjusted for chronological age (see Methods) is shown on the x-axis. For example, CXRs with the ‘fibrosis’ label are likely to be estimated 1.4 years older than the chronological age.”

Reviewer #1, Specific comment #11. Results:

Figure 5 – Please increase the font size for your y-axis labels. Can you clarify what summary statistic you are showing in Parts (a) and (b) ? There is a typo in the title of Part a (“history”). In part (c), does this use individuals of all ages? I wonder if this effect would be even larger if you stratified by chronological age groups. Please add a risk table to the Kaplan-Meier plot to show how many individuals are uncensored at certain timepoints.

Response:

Thank you for this valuable feedback. We also apologize for the typo in the figure and the ambiguity of our plot. We increased the y-axis font and corrected the “history” typo.

In the revised Fig. 5a and b, we show the effect of past clinical history and measurements on X-ray age derived from linear regression analysis. This shows the effect of clinical history or measurements on X-ray age. For example, patients with atrial arrhythmias (atrial fibrillation or atrial flutter) are estimated to be older than their chronological age by 1.2 years.

As the reviewer pointed out, this predictive effect is even stronger in the elderly (HR: 1.019 [95% CI, 1.005–1.032], $P = 6.69 \times 10^{-3}$ vs HR 1.024 [95% CI, 1.01 – 1.038], $p = 9.4 \times 10^{-4}$

(≥ 65 y.o.)). This result is shown in Supplementary Fig. 7. Since the survival plot in the revised fig. 5c shows an adjusted survival curve corrected for conventional risk factors (such as age, sex, and BMI), a censoring table is not available, unlike a usual Kaplan-Meier curve. We revised our figure and figure legends as follows.

Fig. 5 Relationship between X-ray age and clinical characteristics and outcome in heart failure patients.

Past clinical history (**a**) and continuous clinical measurements (**b**) affected X-ray age. The effect of specific clinical history or clinical measurements on the predicted age is shown on the x-axis with a 95% confidence interval. For example, the X-ray age of patients

with "atrial fibrillation or atrial flutter" is likely to be estimated 1.22 years older than their chronological age. HTN, hypertension; DM, diabetes mellitus; DL, dyslipidemia; HUA, hyperuricemia; AFAFL, atrial fibrillation or atrial flutter; COPD, chronic obstructive pulmonary disease; device, cardiac pacemaker, implantable cardioverter defibrillator, or cardiac resynchronization therapy devices LAD, left atrial diameter; LVEF, left ventricular ejection fraction; LVDd, left ventricular end-diastolic diameter; TC, total cholesterol; BS, blood sugar (glucose); HR, heart rate; dBP, diastolic blood pressure; sBP, systolic blood pressure. **c**, Adjusted event-free survival curve for heart failure patients stratified by the age discrepancy between the chronological age and X-ray age. Event was defined as the composite endpoint of heart failure re-hospitalization, heart transplantation, and all-cause mortality. The top 20% of patients, middle 60%, and bottom 20% were grouped as older, middle, and younger, respectively. **d**, Additional value of age discrepancy as assessed by the paired difference of risk scores derived from the Cox proportional hazard model. Figure shows the empirical distribution function of the change in estimated risk score for heart failure patients in the model of conventional risk factors (age, sex, body mass index, LVEF, NT-proBNP, hemoglobin, and estimated glomerular filtration rate) without age discrepancy (thick solid line) and the model with age discrepancy (thin blue solid line). The difference between the areas under the two curves is IDI, and the distances between two black dots and between two gray dots are cNRI and median improvement, respectively.

Reviewer #1, Specific comment #12. Results:

Supplementary Figure 2 - For the Heart failure dataset it is unclear why these chest x-rays were obtained. Are these admission CXRs? Were they obtained for any particular indication during stay?

Response:

We thank the reviewer for this important comment. The admission CXRs were used for the dataset, as stated in the method section, since heart failure patients had routine CXRs at admission.

Lines 412-413 "Frontal CXRs within 2 days of hospital admission were used in the analysis."

Reviewer #1, Specific comment #13. Results:

For the MIMIC dataset, no cohort characteristics were provided. Do you have demographic/co-morbidity data for this cohort?

Response: We thank the reviewer for this important suggestion, which we agree with. We constructed a demographic table of the MIMIC dataset used in this study (Supplementary Table 11)

Supplementary Table 11. Characteristics of patients in the MIMIC database

clinical measurements (unit)	(n=3586)
-----------------

Age (years old)	71 [61, 80]
The number of males (%)	2097 (58.5)
Height (cm)	170 [163, 178]
Weight (kg)	80.6 [67.7, 95.95]
BMI (kg/m ²)	28.1 [24.5, 32.8]
Medical history	
myocardial infarction	1666 (46.5)
congestive heart failure	2788 (77.7)
peripheral vascular disease	624 (17.4)
chronic pulmonary disease	1320 (36.8)
cerebrovascular disease	384 (10.7)
diabetes	1480 (41.3)
Systolic BP (mmHg)	114 [105, 125]
Diastolic BP (mmHg)	61 [54, 68]
Heart rate (/min)	83 [73, 95]
SpO ₂ (%)	97 [95, 98]
Respiratory rate (/min)	20 [17, 22]
Laboratory measurements	
Hb (g/dl)	11.4 [9.8, 13.0]
Hct (%)	35.0 [30.6, 39.9]
WBC (10 ³)	12.60 [9.0, 17.4]
PLT (10 ³)	209 [158, 273]
Na (mEq/L)	139 [137, 142]
K (mEq/L)	4.5 [4.11, 5.0]
T-Bil (mg/dl)	0.7 [0.4, 1.2]
AST (U/L)	45 [26, 105]
ALT (U/L)	30 [18, 66]
ALP (U/L)	87 [65, 124]
LDH (U/L)	302 [218, 494.5]
Alb (mg/dl)	3.4 [2.9, 3.8]
CRP (mg/L)	71.30 [24.57, 147.55]
Cre (mg/dl)	1.30 [1.00, 2.10]
eGFR (ml·min ⁻¹ ·1.86 m ⁻²)	51.33 [29.69, 74.73]
Glucose (mg/dl)	134 [115, 169]
TP (mg/dl)	6.0 [5.4, 6.4]

Characteristics of cardiovascular disease patients in MIMIC database. Continuous variables are presented as medians [interquartile ranges]. Categorical variables are presented as n (%). BMI, body mass index; BP, blood pressure; Hb, hemoglobin; Hct, hematocrit; WBC, white blood cell count; PLT, platelet count; T-Bil, total bilirubin; AST, aspartate aminotransferase; ALT, alanine aminotransferase; ALP, alkaline phosphatase; UA, urinary acid; CRP, C-reactive protein; Cre, creatinine; eGFR, estimated glomerular filtration rate; Lymph, lymphocytes; BS, blood glucose; TP, total protein

Reviewer #1, Specific comment #14. Discussion:

“Our evaluation method is robust and fair” – I would avoid this language and instead just state that “we evaluated the estimation performance on an external..”

Response:

We thank the reviewer for pointing this out, which we agree with. We modified our manuscript accordingly.

Lines 323-325 “We evaluated the estimation performance on an external test dataset as well as an entirely independent JSRT dataset, neither of which were used during the training phase.”

Reviewer #1, Specific comment #15. Discussion:

Line 346-347 The Lu method predicts all-cause mortality, not “mortality risk of lung cancer” – please correct

Response: We thank the reviewer for pointing this out. We modified our manuscript accordingly.

Lines 331-333 “Therefore, Raghu et al.’s method is more similar to Lu et al.’s method⁵⁴, which is a model for predicting all-cause mortality from CXR in lung cancer cohort.”

Reviewer #1, Specific comment #14. Discussion:

Line 348-349 The Raghu paper uses CXR-14 and other public cohorts for pre-training – I wouldn’t state that the training dataset here is larger.

Response:

We thank the reviewer for pointing this out, which we agree with. We modified our manuscript accordingly.

Lines 333-335 “Compared to their model, our DNN model is more straightforward in that our DNN is directly trained to estimate patient’s chronological age from CXR.”

Reviewer #1, Specific comment #15. Discussion:

Line 364-365 “X-ray age can be a simple health indicator that reflects the aging state of the heart and vessels” – It’s unknown whether the model is using the heart/vessels to make an age estimation. The black-box nature of deep learning makes it difficult to conclude this.

Response:

We thank the reviewer for this comment and agree that this may be too strongly worded. We revised our manuscript as follows.

Lines 349-351 “Our results suggest that X-ray age has the potential to be used as a simple health indicator, which estimates possible diseases affecting the heart and vessels.”

Reviewer #1, Specific comment #16. Discussion:

Line 373-376 the phrase “in clinical practice” is repeated twice.

Response:

We sincerely apologize for this typographical error and thank the reviewer for pointing this out. We have fixed this.

Lines 359-362 “In clinical practice, “lung age” estimated from spirometry forced expiratory volume (FEV) ⁶⁴ and “vascular age” estimated from carotid artery ultrasonography ⁶⁵ are used as simple health indicators, and these methods help clinicians explain test results to patients.”

Reviewer #2 (Remarks to the Author):

Overall, this is a well prepared manuscript which utilizes multiple datasets to develop sex and age prediction models based on chest radiographs and correlate prediction discrepancies with cardiovascular disease risk. This is an interesting strategy for health risk assessment and the conclusions are appropriately caveated with a thorough discussion of limitations. Additionally, the supplementary material is thorough and appropriate.

Response: We are grateful for the reviewer's insightful feedback evaluation of our manuscript. We sincerely appreciate the reviewer's opinion. We have addressed the reviewer's concerns, performed additional analyses, and revised our manuscript accordingly.

The following are our point-by-point responses to each of the reviewers' comments.

Reviewer #2, Major comment #1:

1. Consider citing additional references on deep learning and estimation of patient age and sex from chest radiographs:

Sabottke CF, Breaux MA, Spieler BM. Estimation of age in unidentified patients via chest radiography using convolutional neural network regression. *Emerg Radiol.* 2020 Oct;27(5):463-468. doi: 10.1007/s10140-020-01782-5. Epub 2020 Apr 28. PMID: 32347410.

Yi PH, Wei J, Kim TK, Shin J, Sair HI, Hui FK, Hager GD, Lin CT. Radiology "forensics": determination of age and sex from chest radiographs using deep learning. *Emerg Radiol.* 2021 Oct;28(5):949-954. doi: 10.1007/s10140-021-01953-y. Epub 2021 Jun 5. PMID: 34089126.

Yang CY, Pan YJ, Chou Y, Yang CJ, Kao CC, Huang KC, Chang JS, Chen HC, Kuo KH. Using Deep Neural Networks for Predicting Age and Sex in Healthy Adult Chest Radiographs. *J Clin Med.* 2021 Sep 27;10(19):4431. doi: 10.3390/jcm10194431. PMID: 34640449; PMCID: PMC8509558.

Response:

We thank the reviewer for listing these recent publications related to age estimation from CXRs. We have included these references in our manuscript.

Lines 75-79 “Because aging²¹ and sex difference²² change radiological findings of CXR, several studies have explored estimating a patient’s age from CXR and developing artificial intelligence capable of conducting the task²³⁻²⁸. However, the estimation accuracy of those models has not been validated with independent external test data^{24, 26-28}.”

Lines 316-317 “Although aging is associated with CXR findings, few studies have reported age estimation from CXR images²³⁻²⁸.”

Reviewer #2, Major comment #2:

2. Line 367: “However, it is not easy to take facial photographs of patients in clinical settings due to privacy concerns, which hinders the clinical application of perceived age.” This is not necessarily true. Some healthcare institutions will include photographs of patient’s faces in the electronic health record for almost all patients. However, such a dataset would not likely be readily released for research purposes. This statement should perhaps be caveated or modified.

We thank the reviewer for this important suggestion, which we agree with. We modified our manuscript as follows.

Lines 353-355 “However, the combined dataset of patient facial photographs and clinical information would not be available for research purposes due to ethical and privacy concerns, which hinders the clinical application of perceived age.”

Reviewer #2, Major comment #3:

3. After reviewing figure 4, I don't think the discussion which begins on line 393 is necessarily correct:

“Fourth, as described above, an older estimated age does not necessarily mean worse CXR findings. For instance, CXRs with findings of consolidation or effusion were estimated to be younger than the actual age. The transparency of the lung field on CXR increases with age. This is caused by a decrease in the thickness of the chest wall in elderly people . Therefore, we speculate that patients with effusion or consolidation findings could potentially be estimated as younger than their actual age because these findings reduce the transparency of the lung field. This fact implies that, as we demonstrated ,not estimated age itself but the dissociation between actual age and estimated age may be important for detecting the presence of a disease.”

This speculation does not make sense to me because consolidation and effusion are localizable opacities while the increased transparency associated with aging would impact the bilateral lung fields diffusely. The neural networks should not be conflating effusions and consolidations with diffuse changes in penetration associated with differences in chest wall thickness. My speculation is that effusion and consolidation lead to younger predictions based on a bias in the clinical reason for obtaining chest radiographs in young patients. Younger patients in a hospital setting may be less likely to be imaged with a chest radiograph if they do not have cardiopulmonary symptoms. However, pneumonia, consolidation, or effusion may warrant repeated imaging. I think that a supplementary figure or table may be helpful to clarify this issue. Perhaps a figure can be constructed which shows the number or percentage of cases with various findings such as consolidation or effusion broken down based on 5 year intervals of patient age. This would be helpful to better understand the data being presented in figure 4e.

Response: We thank the reviewer for raising this important point, providing these specific suggestions, and the reviewer's speculation regarding the reasons for this result. We agree that it is counterintuitive that “consolidation” and “effusion” lead to a younger estimation.

To further explore this result, as per the reviewer’s feedback, we constructed a table and figure showing the percentage of cases with different finding labels broken down based on 5-year intervals of patient age (**response Table and Fig.** below). The figure shows that the proportion of CXRs with the “consolidation” label was greater in younger people, while CXRs with the “effusion” label were greater in older people. However, the data included multiple CXRs per patient, which might lead to some bias. For example, patients with “consolidation” due to pneumonia would take multiple CXRs for follow-up, as the reviewer suggested. We subsequently removed the duplicated CXRs (multiple CXRs per patient) and conducted the analysis. As **Fig. 4d** below shows, “fibrosis” and “effusion” were associated with older X-ray age estimation, while no label was associated with younger X-ray age estimation. We truly appreciate the reviewer’s careful assessment and useful suggestion to re-analyze our results, which has improved our manuscript.

Please also refer to the response to **Reviewer #1, Specific comment #7; Results; and Reviewer #3, Minor comment #3.**

Response Table. Frequency of finding labels in the different age groups

age_group	n	cardiomegaly	atelectasis	pneumothorax	emphysema	nodule	fibrosis	effusion	consolidation	pneumonia
(0,5]	396	1.77%	6.31%	0.51%	0.00%	4.29%	0.25%	1.77%	26.01%	3.28%
(5,10]	1246	2.89%	6.18%	3.29%	0.16%	4.98%	0.16%	8.75%	29.70%	2.09%
(10,15]	1926	4.31%	6.23%	6.91%	1.35%	7.11%	0.31%	9.55%	23.05%	2.02%
(15,20]	3538	1.53%	6.61%	7.43%	3.22%	7.46%	0.82%	7.63%	20.97%	1.27%
(20,25]	5717	1.89%	7.33%	5.56%	1.96%	8.29%	0.68%	8.97%	23.30%	1.70%
(25,30]	6372	2.68%	7.36%	4.41%	1.51%	9.37%	1.16%	11.32%	24.76%	1.41%
(30,35]	7790	2.75%	7.50%	4.60%	1.86%	7.70%	0.95%	11.67%	23.68%	1.57%
(35,40]	7537	2.27%	8.41%	3.68%	1.17%	9.27%	1.13%	8.65%	20.62%	1.43%
(40,45]	9173	1.94%	9.86%	4.28%	1.21%	10.24%	1.31%	10.95%	21.27%	1.08%
(45,50]	10823	2.30%	10.36%	5.08%	1.92%	9.80%	1.88%	10.39%	20.68%	1.07%
(50,55]	12569	2.46%	11.75%	4.18%	2.32%	10.95%	1.78%	12.36%	20.73%	1.05%
(55,60]	12348	2.53%	12.59%	4.88%	2.58%	11.62%	1.62%	14.83%	21.91%	1.24%
(60,65]	9853	2.34%	12.06%	5.48%	3.61%	11.84%	1.79%	13.22%	20.16%	1.21%
(65,70]	6700	2.46%	12.64%	4.67%	2.88%	11.09%	2.36%	13.94%	22.94%	1.49%

(70,75]	3593	2.62%	14.50%	4.68%	3.45%	9.30%	2.37%	16.25%	23.04%	0.95%
(75,80]	1634	4.16%	12.18%	4.96%	4.10%	10.89%	3.24%	16.28%	19.52%	1.04%
(80,85]	567	4.41%	11.99%	7.41%	4.59%	10.93%	4.23%	13.40%	22.05%	1.41%
(85,90]	212	5.19%	15.57%	0.47%	2.83%	16.04%	2.36%	19.34%	27.83%	0.94%
(90,95]	35	2.86%	17.14%	0.00%	0.00%	17.14%	5.71%	8.57%	14.29%	0.00%

Response Fig. The proportion of finding labels in the different age groups.

Lines 202-204 “Regarding each finding label, CXRs with findings of lung fibrosis and effusion were estimated to be significantly older (fibrosis: + 1.41 [0.17-2.66] years; effusion: +0.81 [0.10 – 1.52] years) than their chronological age (Fig. 4d).”

Lines 513-515 “To analyze the association between X-ray age and finding labels, linear regression was performed using the validation and test data. Only the first CXR was used for analysis for patients with more than one CXR.”

Fig. 4 Characteristics of images resulting in inaccurate age estimation by the deep learning model.

Reviewer #3 (Remarks to the Author):

In this paper, the authors present a deep learning based computer vision system for estimating “chest X-ray age” from frontal chest radiographs and show that this “chest X-ray age” is predictive of clinically meaningful outcomes and that it is additive to traditional risk factors. The authors also present a model to predict sex based on chest radiographs, but this application is far less interesting/clinically applicable and seems to detract from the main thesis of the paper.

There are many strengths of this paper – it is methodologically sound and crucially, it was validated/tested on a number of external datasets demonstrating its generalizability to different patient cohorts which is so important in an era where awareness of the pitfalls of dataset shift is growing. This sets this work apart from prior similar efforts. Additionally, the fact that the authors’ model is capable of accurately estimating patient age on chest radiographs, a task that the authors demonstrate eludes human interpreters in their comparison to a consensus of experienced X-ray interpretations, is a fascinating example of the power of deep learning technologies to unlock hidden patterns in complex data.

Response: We appreciate the reviewer’s careful evaluation of the strength of our work. We have addressed the reviewer’s concerns, performed additional analyses, and revised our manuscript accordingly.

The following are our point-by-point responses to each comment.

Reviewer #3, Major comment #1:

Despite the strengths of this work as noted above, I do have concerns about the overall premise of the paper – namely that the discrepancy between predicted age by the deep learning model and actual age is predictive of clinically meaningful outcomes. Each of the CNN architectures explored in this paper were trained using the label of actual chronological age – therefore the difference between predicted age and actual age is essentially a measure of the error of the model. I understand the author’s reasoning that this may be a clinically meaningful metric – in an appropriately fit model that fits to features in the chest radiograph that might suggest an older age, then the model might have a larger error of prediction on those chest radiographs that have more of these features, suggesting a worse clinical prognosis. However, I have significant concerns about this assumption.

First, this premise is highly contingent on the training process itself and the dataset on which the model was trained. As an example, if one were to be able to theoretically train a nearly perfectly performing model (for example using a different architecture) while still not sacrificing generalizability to external datasets, then this discrepancy between predicted age and actual chronological age would disappear. In this case, the better performing model might fit to more robust features in the chest X-ray that are associated with chronological age regardless of the pathology present on chest X-ray. It seems that in this case the authors were fortunate to train a model that seemed to fit to features associated with pathology on the chest radiographs as they demonstrated in their analysis of the association between chest X-ray/chronological age discrepancy and pathological labels in the NIH CXR dataset.

The frailty of this assumption is also demonstrated by the variable performance of the model on the various datasets under study in this paper. For example, the performance of the age prediction model in the HF dataset was much lower than in other datasets and the authors theorize that this may be because there is more pathology present in

this dataset and thus greater discrepancy between the model's predicted age and actual age. However, what would happen if the model was TRAINED on the HF dataset? In minimizing loss/error on this dataset, the model would likely learn to ignore some of this pathology present on chest radiographs and instead focus on other features that might be associated with age.

Expanding further on this point, I am unsure about the clinical utility of a model that predicts "chest X-ray age", when this model may essentially be fitting to pathological features present on chest radiographs. Would a model that is trained to directly identify pathology on chest radiographs be a stronger predictor of outcomes?

In order to prove that "chest X-ray age" is clinically useful beyond identifying pathological features present on chest radiographs, my recommendation is that the authors train a separate model to identify pathologic labels (e.g. cardiomegaly, pleural effusion, etc.) on the NIH CXR dataset. While evaluation of the accuracy of these identified labels might not be possible on some of the external datasets due to lack of ground truth labels, the authors can still explore whether these automatically generated labels might be predictive of primary composite outcomes (e.g. composite outcome of HF hospitalization or mortality in the HF cohort). The authors could explore the additive predictive value of individual pathological labels and/or ANY pathological label in multivariate modeling. The authors could also explore whether "chest X-ray age" retains significance in a multivariate Cox model adjusted for these labels or that introducing chest X-ray age into a model adjusted for these labels still improves AIC/BIC in order to definitively conclude that "chest X-ray age" is a clinically meaningful metric.

Response: We appreciate the reviewer's insightful comments and constructive suggestions. As the reviewer indicated, we need to consider whether this model may inherently fit the pathological features present in CXR, or whether our x-ray age is useful for our purposes, such as disease prognostic stratification, even after correcting for the effects of pathological features. Thus, to demonstrate the usefulness of X-ray age as a prognostic marker beyond identifying finding labels from CXR, we have included the CXR abnormality information in

the Cox model and investigated the performance improvement by addition of age discrepancy under the presence of CXR abnormal finding label. The results showed that the addition of age discrepancy improved the model performance, even when CXR abnormality labels were included in the model. These results suggest that our x-ray age model would be a clinically meaningful measure independent of CXR finding labels.

Lines 541-549 “To assess whether X-ray age is clinically useful beyond simply identifying pathological features on CXRs, we also included the CXR abnormality information in the Cox model, using a recently developed abnormality classification DNN⁷⁴. The model output is binary values of whether is normal (0) or abnormal (1) CXR, and its probability. We calculated the abnormality binary value and its probability from the heart failure cohort’s CXRs. We compared Cox model results, including the binary value (abnormality) or logit [$\text{logit}(p) = \log \left(\frac{p}{1-p} \right)$] of probability (abnormal score) with and without X-ray age information (**Supplementary Table 10**).”

Supplementary Table 10. Comparison of different Cox proportional hazards models and improved predictive performance due to addition of age discrepancy to models

Model	Covariates in Cox model	AIC	Compared with	Performance improvement											
				IDI				cNRI				MI			
				Estimate	lower 95%	upper 95%	P value	Estimate	lower 95%	upper 95%	P value	Estimate	lower 95%	upper 95%	P value
Model 1	Age + Sex + BMI + LVEF +log(NT-proBNP) + Hb + eGFR	7608.6													
Model 2	Age + Sex + BMI + LVEF +log(NT-proBNP) + Hb + eGFR + Age_discrepancy	7206.6	vs Model 1	0.0104	0.0007	0.0245	0.020	0.134	0.025	0.201	0.010	0.0110	0.0008	0.0250	0.020
Model 3	Age + Sex + BMI + LVEF +log(NT-proBNP) + Hb + eGFR + CXR_Abnormality	7204.4													
Model 4	Age + Sex + BMI + LVEF +log(NT-proBNP) + Hb + eGFR + CXR_Abnormality + Age_discrepancy	7198.9	vs Model 3	0.0105	0.0007	0.0251	0.030	0.135	0.034	0.202	0.030	0.0110	0.0008	0.0260	0.030
Model 5	Age + Sex + BMI + LVEF +log(NT-proBNP) + Hb + eGFR + CXR_Abnormality_score	7206.1													
Model 6	Age + Sex + BMI + LVEF +log(NT-proBNP) + Hb + eGFR + CXR_Abnormality_score + Age_discrepancy	7200.4	vs Model 1	0.0105	0.0008	0.0250	0.020	0.121	0.036	0.197	0.020	0.0120	0.0008	0.0250	0.020

Comparison of different Cox proportional hazards models with different variables and the additional value of age discrepancy as assessed by the paired difference of risk scores derived from the Cox proportional hazard model is shown. Improvement in the predictive performance of the Cox model assessed by Akaike's Information Criterion (AIC), integrated discrimination improvement (IDI); continuous net reclassification improvement (cNRI), and median improvement (MI).

BMI, body mass index; LVEF, left ventricular ejection fraction; Hb, hemoglobin; eGFR, estimated glomerular filtration rate; X-ray age, age estimated using the deep learning model; Age_discrepancy, the difference between X-ray age and chronological age (X-ray age - chronological age); CXR_abnormality, binary variable for whether CXR is abnormal or not from abnormality detection deep learning model; CXR_abnormality_score, continuous variable indicating whether there are abnormalities in CXR from abnormality detection deep learning model.

Reviewer #3, Minor comment #1:

1. As above, I am unsure what the sex prediction model adds to this paper. This model does not appear to be clinically meaningful and sex prediction does not appear to be a very interesting task as the authors demonstrate that this model seems to fit to breast shadows on the chest radiograph. My suggestion would be to remove this model completely from the paper as it detracts from the study's main message.

Response: We thank the reviewer for this important feedback. We agree that the most meaningful part of our study is age estimation from CXR and its clinical utility. Accordingly, we have removed the sex estimation part from our manuscript.

Reviewer #3, Minor comment #2:

Lines 118-119 – Why not use an ensemble of all models together rather than choosing the best performing model? There is evidence that model ensembling can help resolve the bias-variance trade-off and may lead to better generalizability across diverse datasets. You could compare performance of an ensemble of all models (or X top performing models) vs each individual model to explore this possibility.

Response:

We thank the reviewer for this important suggestion. We agree that an ensemble prediction model could improve estimation performance. Therefore, we compared the ensemble of 11 different deep learning models with a single model (SENet154, **Supplementary Table 6**). Consequently, the ensemble prediction model showed comparable estimation performance in the JSRT dataset compared to the SENet154-based model, whilst it showed no statistically significant improvement. Therefore, the SENet154-based model was used in subsequent analyses.

We added Supplementary Table 6 and modified our manuscript accordingly.

Lines 154-159 “We further compared the ensemble inference of eleven different architectures of DNNs with the individual SENet154 model, and found no significant difference in prediction performance on either the test or JSRT data (Pearson’s r: 0.962 vs 0.960, p=1; ICC: 0.957 vs 0.955, p = 1 in the test data; Pearson’s r: 0.916 vs 0.912, p=1; ICC: 0.878 vs 0.889, p = 0.053 in the JSRT data, **Supplementary Table 6**).”

Lines 510-513 “For the ensemble model, estimated age of the 11 different DNN models was averaged and compared with the SENet154-based single model output. The P value was derived using 20,000 bootstrap replications method (**Supplementary Table 6**).”

Supplementary Table 6. Estimation performance of a single DNN model and an ensemble model.

Dataset	Test dataset			JSRT dataset		
	SENet154 model	Ensemble prediction model	P value	SENet154 model	Ensemble prediction model	P value
R	0.962	0.960	1	0.916	0.912	1
MAE	3.666	3.711	0.558	4.953	4.709	0.075
ICC	0.957	0.955	1	0.878	0.889	0.053

Estimation performance of SENet154-based single DNN model and the ensemble estimation model of 11 different DNN architectures.

The P value was derived using 20,000 bootstrap replications. R, Pearson's r between the chronological and estimated age; MAE, mean absolute error; ICC, intraclass correlation coefficient.

Reviewer #3, Minor comment #3:

The idea that consolidation and effusion were associated with younger CXR age than chronological age is counterintuitive to me. I appreciate the authors' attempt at explaining this phenomenon in the discussion, however, one would think that there are enough examples of consolidation or effusion in the NIH CXR dataset such that the model could learn to differentiate focal opacities associated with effusion or consolidation vs. generally lower transparency of the lung fields in younger patients. This explanation is therefore not very satisfying for an intriguing finding. Perhaps, the authors could provide a more in depth analysis of visualization techniques such as Grad-CAM and guided backprop on some examples of these chest radiographs in order to explore which features the model might be using to predict a lower "chest-Xray age".

Response:

We thank the reviewer for raising this key question. We also agree that this result is counterintuitive.

In our original analysis, the association between finding labels and X-ray age was analyzed using both validation and test data. However, the data included multiple CXRs per patient, which resulted in some bias. For example, patients with consolidation due to pneumonia would take multiple CXRs for follow-up. We have performed additional analysis by removing those duplicates. Please also refer to the response to **Reviewer #1, Specific comment #7. Results and Reviewer #2, Major comment #3.**

As the reviewer suggested, we also performed Grad-CAM visualization of CXRs. The heatmap highlighted a larger area in CXRs with effusion and consolidation, compared to "no finding" CXRs. Unfortunately, this result does not allow us to conclude that the highlighted area in the heatmap is a determining factor for younger or older X-ray age; however, it suggests that the "Effusion" and "Consolidation" findings may have some influence on the X-ray age.

Response Fig. Grad-CAM and guided-Grad-CAM visualization of CXRs with "no findings", "Effusion" and "Consolidation"

Reviewer #3, Minor comment #4:

In the multivariable Cox models, there appears to be a difference between the nominal models used to evaluate the significance of CXR age and Age discrepancy (Supplementary table 12). Why was chronological age left out of the baseline model for CXR age? Please provide an additional multivariable model analyzing the significance of CXR age in a multivariable model adjusted for all of the baseline variables including chronological age.

Response: We thank the reviewer for raising this important point. Since age discrepancy was calculated by the formula [X-ray age – chronological age], we thought that a model including both X-ray age and the age discrepancy was enough to address the significance of X-ray age. i.e. the model contained covariates derived from both X-ray and chronological ages. We therefore removed chronological age from the baseline model when comparing two models with and without X-ray age. On the other hand, to directly address the reviewers' concern, we tested whether adding X-ray age to the model, in addition to chronological age, improved the performance. As shown in the Response table below, we evaluated the performance improvement of the model by adding X-ray age by checking multiple indices such as AIC, IDI, cNRI, and MI, and found a statistically significant improvement.

Response table. Cox proportional hazard model for heart failure prognosis with or without X-ray age information

Covariates	AIC	Performance improvement					
		IDI		cNRI		MI	
	Estimate (95% C.I.)	P value	Estimate (95% C.I.)	P value	Estimate (95% C.I.)	P value	
Age + Sex + BMI + LVEF + log(NT-proBNP) + Hb + eGFR	7608.6						
Age + Sex + BMI + LVEF + log(NT-proBNP) + Hb + eGFR + X-ray age	7206.6	0.0104 (0.0014 - 0.026)	0.010	0.134 (0.043 - 0.215)	0.010	0.011 (0.002 - 0.028)	0.010

Reviewer #3, Minor comment #5:

Code availability – it seems a shame that the code is available “on request”. It doesn’t seem that there are proprietary concerns with this model (given the authors used publicly available CNN architectures) and the model was trained on publicly available datasets. For the benefit of the scientific community and to allow external validation of this model across diverse datasets by interested scientists, it is my opinion that both the code to generate this model and the trained weights of the best performing model(s) should be made publicly available on a platform such as Github, Bitbucket, or Sourceforge if the paper is to be accepted for publication.

Response: We thank the reviewer for this valuable suggestion. Accordingly, we decided to release both our code and trained model for X-ray age estimation on GitHub (http://github.com/pirocv/xray_age). It will be publicly available as soon as our manuscript is accepted for publication. We have changed the data availability statement accordingly.

Lines 567-568 “Both our code and trained model for estimating x-ray age on GitHub (http://github.com/pirocv/xray_age).”

Reviewers' comments:

Reviewer #1 (Remarks to the Author):

The authors present an X-ray Age based on deep learning applied to chest x-ray images trained using >100,000 images. The X-ray age was applied to an external validation population of 1,562 consecutive patients hospitalized with heart failure and 3,586 with cardiovascular disease admitted to the ICU. Elevated X-ray age was associated with worse clinical outcomes in both populations, suggesting that the discrepancy between x-ray age and chronologic age serves as an indicator of cardiovascular abnormalities (though the effect sizes in this study are rather small). Overall a nice study that adds to the growing body of literature that deep learning applied to chest x-rays can lead to useful prognostic biomarkers.

Thank you to the authors for your thorough revisions. The manuscript is much improved; however, I still have some minor concerns

Specific Comments:

Introduction

Lines 51-53 “therefore, variation in perceived age may increase. Perceived age is also difficult to implement in actual clinical practice because it is difficult to obtain decisions by more than 10 medical professionals.”

Just because 10 medical professionals were used in these studies, doesn't mean that you need 10 professionals to estimate a patient's perceived age. Perceived age is used all the time by medical professionals during routine care. Maybe it would be more accurate to say that there haven't been studies that have tested whether perceived age is a robust predictor of age-related disease.

Lines 79-82 The other studies you cited have already shown that chest x-ray age is associated with disease after adjusting for prognostic factors, and these have been validated on independent testing data. Please clarify your added contribution. The novelty in your paper appears to be the specific use case showing that the chest x-ray age is associated with poor prognosis in patient's with heart failure/CVD instead of in the general population (which is what the other chest x-ray works have shown using cancer screening trial data)

Lines 80-83 “Although an association between estimated age and prognosis has been suggested in the lung cancer patients cohort, it is still unclear whether there is an association between estimated age and cardiovascular disease and its prognosis, or whether estimated age has any additional value after adjusting for other prognostic factors.”

The revision has still not alleviated my concern about prior work. Others (Raghu et al) have already shown an association between estimated age and cardiovascular mortality, and have shown that this signal is robust to adjustment for other prognostic factors. What is the added value of this study? The ability to predict heart failure using estimated age – chronological age is new.

Results

Lines 106-108 “After removing the age outliers (since they are considered mislabeled, see Methods for details), 63,328 (56%) of the 112,104 X-rays were male CXRs.”

The phrase “age outliers” is still vague to me. Could you just say “after removing individuals with age

> 140 years...”

Lines 148-150 “The model trained with the entire CXR image showed better estimation performance compared to the DNN model trained on CXRs labeled as “No Finding” (Pearson’s r: 0.962 vs 0.951, $p < 0.0001$; ICC1: 0.957 vs 0.945, $p = 0.0005$).”

Please change “with the entire CXR image” to “with all CXR images”. As written, this sounds like you are using the full image instead of cropping out a certain region of the image.

Lines 223-228 “Although the performance of age estimation was expected to decrease because all the CXRs were of HF patients and accordingly had some abnormal findings, there was still a significant positive correlation between estimated and chronological age (Pearson’s r: 0.769 [95% CI, 0.747–0.789], $p = 4.6 \times 10^{-291}$; ICC1 0.257 [95% CI, 0.218– 0.296], $p = 2.1 \times 10^{-25}$). This result suggests that our DNN is still able to estimate aging in the patients with a disease, although its accuracy is reduced.” Line 244 There is a major confounder here that you are examining associations with co-morbidities in patients who are hospitalized with heart failure. Please include this as a limitation in the discussion.

It is still unclear how many patients had repeat imaging to calculate the ICC in the heart failure cohort.

Discussion

Lines 330-331 Their model was trained to predict this “biological age” in cohorts consisting of patients with cancer and heavy smokers

The cohorts in the Raghu et al paper were not only patients with cancer/heavy smokers. This was a cancer screening trial of asymptomatic individuals. I would delete these phrases from the discussion.

Lines 331-333 “Therefore, Raghu et al.’s method is more similar to Lu et al.’s method, which is a model for predicting all-cause mortality from CXR in lung cancer cohort.”

The Lu model was not trained in a “lung cancer cohort”. This was a general cancer screening trial – I would just delete this phrase.

Reviewer #3 (Remarks to the Author):

The authors have done an excellent job thoroughly responding to all of the reviewer comments. The result is a very strong manuscript and important contribution to the literature. In particular, the realization that the association between consolidation/effusion and younger CXR age was due to bias in including multiple CXRs per patient and after correction showing that lung fibrosis is associated with older CXR age (a result that is biologically plausible) is a shining example of the utility of the peer review process. Additionally, the focus on CXR age alone (removing the parts pertaining to sex estimation) and the thorough demonstration that CXR age has prognostic significance beyond both clinical characteristics and abnormalities seen on CXR using a DNN classifier (and outperforms estimations made by clinicians) makes this quite a fascinating addition to the body of evidence

showing the remarkable ability of DL to reveal important features not readily apparent to even expert reviewers.

I found the author's response to minor comment #3 particularly interesting, given the gradCAM heatmaps presented seem to highlight features in the superior mediastinum/sternum, as well as bony features of the clavicles and shoulders (regardless of pathology present in the CXR). If possible, I would encourage the authors to include a figure showing sample GradCAM heatmaps either in the main manuscript or the supplemental materials and comment on this. Would also be interesting to compare to any other age prediction models trained on CXRs that may have published saliency maps to see if they fit to similar features. My sense is that the model is fitting to differences in bony structures (e.g. reduced density indicating osteoporosis or joint spaces indicating osteoarthritis). This of course can not be proven given available data, but is an interesting hypothesis provoking finding.

Minor comment:

Supplementary table 5 - I believe there is a typo. Last row "compared with" column should be "vs Model 5".

Dear Editors and Reviewers,

We sincerely thank all the editors and reviewers for investing a considerable amount of time to provide these high-quality comments, which have helped us to strengthen our study. We are also grateful for your evaluation that our study has been substantially improved and for the opportunity to further revise and resubmit our manuscript.

Based on the reviewers' comments, we modified several sentences to properly reference previous works and have emphasized our new finding on X-ray age, i.e., that the difference between estimated age and chronological age was prognostic, especially in the heart failure population. Additionally, we added discussion and figures to suggest which parts of the Grad-CAM heatmap for X-ray age estimation were of interest (Fig. 3. and Supplementary Fig. 6). We also fixed the typos.

As described in “the response to the reviewers' letter” of the 1st revision, our code and trained model to estimate X-ray age will be publicly available on http://github.com/pirocv/xray_age as soon as it gets accepted for publication.

We hope you will find that these revisions have made our manuscript acceptable for publication in *Communications Medicine*. The point-by-point responses to all comments from the reviewers are given below.

REVIEWER COMMENTS

Reviewer #1 (Remarks to the Author):

The authors present an X-ray Age based on deep learning applied to chest x-ray images trained using >100,000 images. The X-ray age was applied to an external validation population of 1,562 consecutive patients hospitalized with heart failure and 3,586 with cardiovascular disease admitted to the ICU. Elevated X-ray age was associated with worse clinical outcomes in both populations, suggesting that the discrepancy between x-ray age and chronologic age serves as an indicator of cardiovascular abnormalities (though the effect sizes in this study are rather small). Overall a nice study that adds to the growing body of literature that deep learning applied to chest x-rays can lead to useful prognostic biomarkers.

Thank you to the authors for your thorough revisions. The manuscript is much improved; however, I still have some minor concerns.

Response: We thank the reviewer for the careful assessment and apologize for unclear points and ambiguous sentences in the previous manuscript. We also appreciate your additional advice and comments to enhance the clarity and validity of our study. Accordingly, we have revised our manuscript again, especially attempting to clarify the additional contributions of this study. The following is a point-by-point response to each comment:

Reviewer #1, Specific comment #1. Introduction:

Lines 51-53 “therefore, variation in perceived age may increase. Perceived age is also difficult to implement in actual clinical practice because it is difficult to obtain decisions by more than 10 medical professionals.”

Just because 10 medical professionals were used in these studies, doesn't mean that you need 10 professionals to estimate a patient's perceived age. Perceived age is used all the time by medical professionals during routine care. Maybe it would be more accurate to say that there haven't been studies that have tested whether perceived age is a robust predictor of age-related disease.

Response: Thank you for raising this important point. We agree that just because the average age estimated by 10 health care professionals was used in these studies, it would not be necessary to perform the same calculation in a clinical setting, and that medical professionals unconsciously estimate a patient's age from his or her appearance. Accordingly, we revised our manuscript as follows.

(Lines 52-53 in the main manuscript)

“There have been no studies that test whether perceived age is a robust predictor for age-related diseases, including cardiovascular disease.”

Reviewer #1, Specific comment #2. Introduction:

Lines 79-82 The other studies you cited have already shown that chest x-ray age is associated with disease after adjusting for prognostic factors, and these have been validated on independent testing data. Please clarify your added contribution. The novelty in your paper appears to be the specific use case showing that the chest x-ray age is associated with poor prognosis in patient's with heart failure/CVD instead of in the general population (which is what the other chest x-ray works have shown using cancer screening trial data)

Response: Thank you for this important comment. As you pointed out, the association of chest X-ray age with disease has already been reported. In order to clarify our additional contribution to the significance of chest X-ray age estimation, we have

revised the text as follows.

(Lines 79-82 in the main manuscript)

“Although the association of estimated age with disease prognosis has been suggested in the general population of a cancer screening trial cohort ²⁵, it is still unclear whether estimated age can predict prognosis in populations with cardiovascular disease, especially heart failure.”

Reviewer #1, Specific comment #3. Introduction:

Lines 80-83 “Although an association between estimated age and prognosis has been suggested in the lung cancer patients cohort, it is still unclear whether there is an association between estimated age and cardiovascular disease and its prognosis, or whether estimated age has any additional value after adjusting for other prognostic factors.”

The revision has still not alleviated my concern about prior work. Others (Raghu et al) have already shown an association between estimated age and cardiovascular mortality, and have shown that this signal is robust to adjustment for other prognostic factors. What is the added value of this study? The ability to predict heart failure using estimated age – chronological age is new.

Response: We apologize for this ambiguous mention of our study’s added value. Accordingly, we revised the text as follows.

(Lines 82-84 in the main manuscript)

“It also remains unclear whether age discrepancy, i.e. the deviation between chronological and estimated age, has any prognostic value for heart failure.”

Reviewer #1, Specific comment #4. Results:

Lines 106-108 “After removing the age outliers (since they are considered mislabeled, see Methods for details), 63,328 (56%) of the 112,104 X-rays were male CXRs.” The phrase “age outliers” is still vague to me. Could you just say “after removing individuals with age > 140 years...”

Response: Thank you for raising this point and we apologize for the vagueness. Actually, the data included a normal distributed age group (<100 years) and outliers (>140 years). Accordingly, we modified our manuscript as follows.

(Lines 107-109 in the main manuscript)

“After removing individuals with age > 100 years because they were considered mislabeled, 112,104 CXRs remained; of these, 63,328 (56%) were male.”

Reviewer #1, Specific comment #5. Results:

Lines 148-150 “The model trained with the entire CXR image showed better estimation performance compared to the DNN model trained on CXRs labeled as “No Finding” (Pearson’s r: 0.962 vs 0.951, $p < 0.0001$; ICC1: 0.957 vs 0.945, $p = 0.0005$).”

Please change “with the entire CXR image” to “with all CXR images”. As written, this sounds like you are using the full image instead of cropping out a certain region of the image.

Response: We thank the reviewer for this suggestion. Accordingly, we modified our manuscript as follows.

(Lines 149-150 in the main manuscript)

“The model trained with **all CXR images** showed better estimation performance compared to the DNN model trained on CXRs labeled as “No Finding”

Reviewer #1, Specific comment #6. Results:

Lines 223-228 “Although the performance of age estimation was expected to decrease because all the CXRs were of HF patients and accordingly had some abnormal findings, there was still a significant positive correlation between estimated and chronological age (Pearson’s r : 0.769 [95% CI, 0.747–0.789], $p = 4.6 \times 10^{-291}$; ICC1 0.257 [95% CI, 0.218– 0.296], $p = 2.1 \times 10^{-25}$). This result suggests that our DNN is still able to estimate aging in the patients with a disease, although its accuracy is reduced.” Line 244 There is a major confounder here that you are examining associations with co-morbidities in patients who are hospitalized with heart failure. Please include this as a limitation in the discussion.

It is still unclear how many patients had repeat imaging to calculate the ICC in the heart failure cohort.

Response: Thank you for your suggestion on the study limitation. We mention the limitation that we examined associations with co-morbidities in hospitalized patients with heart failure in the discussion section as follows

(Lines 379-380 in the main manuscript)

“**Fourth, we only examined the relationship between the X-ray age, disease and prognosis in patients who were hospitalized with heart failure.**”

As for the number of images to calculate the ICC, we apologize for our insufficient explanation of this point. We used just one

CXR image for each patient (n=1,562) in the heart failure cohort to assess the correlation between chronological age and X-ray age. Thus, no repeated imaging was used. As the reviewer suggested, although ICC is often used for repeated measurements to assess reliability, we calculated ICC not for repeated measurements, but for the strict assessment of two measurements that belonged to the same class (i.e., age vs age). This is because, when X-ray age correlates—but is always estimated lower or higher than chronological age—Pearson’s r value will be still high. Therefore, we think Pearson’s r alone is insufficient for assessing the stringent estimation of performance. ICC refers to correlation within a class of data and it is also used to detect and measure systematic differences between subjects. Using ICC in such a situation had been recommended by the reviewer in the previous revision in Nature Communications.

Reviewer #1, Specific comment #7. Discussion:

Lines 330-331 Their model was trained to predict this “biological age” in cohorts consisting of patients with cancer and heavy smokers

The cohorts in the Raghu et al paper were not only patients with cancer/heavy smokers. This was a cancer screening trial of asymptomatic individuals. I would delete these phrases from the discussion.

Response: We apologize for the inaccurate description of the previous work of Raghu et al. As the reviewer pointed out, these authors did not use only cohorts of cancer/heavy smokers, but a cancer screening cohort including PLCO (asymptomatic individuals aged 55-74), NLST (heavy smoker (30/day) aged 55-74) and public CXR imaging data. Accordingly, we deleted this sentence from our manuscript.

Reviewer #1, Specific comment #8. Results:

Lines 331-333 “Therefore, Raghu et al.’s method is more similar to Lu et al.’s method, which is a model for predicting all-cause mortality from CXR in lung cancer cohort.”

The Lu model was not trained in a “lung cancer cohort”. This was a general cancer screening trial – I would just delete this phrase.

Response: We apologize again for the inaccurate description of their previous work. According to the reviewer’s suggestion, we have deleted this phrase in our revised manuscript.

(Lines 331-335 in the main manuscript)

“In their model development, Raghu et al. defined “biological age” as the patient’s chronological age plus the difference between the expected death age and the actual death age. Compared to their model, our DNN model is more straightforward in that our DNN is directly trained to estimate the patient’s chronological age from a CXR.”

Reviewer #3 (Remarks to the Author):

The authors have done an excellent job thoroughly responding to all of the reviewer comments. The result is a very strong manuscript and important contribution to the literature. In particular, the realization that the association between consolidation/effusion and younger CXR age was due to bias in including multiple CXRs per patient and after correction showing that lung fibrosis is associated with older CXR age (a result that is biologically plausible) is a shining example of the utility of the peer review process. Additionally, the focus on CXR age alone (removing the parts pertaining to sex estimation) and the thorough demonstration that CXR age has prognostic significance beyond both clinical characteristics and abnormalities seen on CXR using a DNN classifier (and outperforms estimations made by clinicians) makes this quite a fascinating addition to the body of evidence showing the remarkable ability of DL to reveal important features not readily apparent to even expert reviewers.

Response: We are grateful for the reviewer's insightful feedback and careful evaluation of our revised manuscript. We also thank the peer review process, which improved our manuscript substantially. The following is a point-by-point response to each of your comments.

Reviewer #2, Major comment #1:

I found the author's response to minor comment #3 particularly interesting, given the gradCAM heatmaps presented seem to highlight features in the superior mediastinum/sternum, as well as bony features of the clavicles and shoulders (regardless of pathology present in the CXR). If possible, I would encourage the authors to include a figure showing sample GradCAM heatmaps either in the main manuscript or the supplemental materials and comment on this. Would also be interesting to compare to any other age prediction models trained on CXRs that may have published saliency maps to see if they fit to similar features. My sense is that the model is fitting to differences in

bony structures (e.g. reduced density indicating osteoporosis or joint spaces indicating osteoarthritis). This of course can not be proven given available data, but is an interesting hypothesis provoking finding.

Response: We thank the reviewer for the interesting and insightful comment on the GradCAM heatmap. We sought to compare our heatmaps to published age prediction models that have saliency maps. As a result, there was a similar tendency for the DNN model to highlight the mediastinum and bony features of the clavicles and shoulders. As the reviewer suggested, we also included heatmap analysis results in the Supplementary Figures and added a comment about them.

(Lines 183-188 in the main manuscript)

“Examples of heatmap images are shown in **Fig. 3** and **Supplementary Fig. 6**. The model mainly focused on the top of the mediastinum and periphery of the rib cage regardless of the pathology present in the CXRs, where the shape and calcification of the aorta seemed to affect the estimation. This pattern is similar to the previously reported CXR-based age prediction model using DNN^{24 25},”

(Supplementary Fig. 6)

Visualization of CXR images with each finding label using Grad-CAM and Guided-back propagation

Example of CXRs with finding labels are shown on left side with the finding labels on top. Heatmap visualization using Grad-CAM (middle) and guided Grad-CAM (right) for each of the CXRs are shown.

**Reviewer #2, Minor comment #2:**

**Supplementary table 5 - I believe there is a typo. Last row "compared with" column should be "vs Model 5".**

**Response:** We apologize for this typo in the Supplementary Table. We have changed "Model 1" into "Model 5" in the last row of Supplementary Table 10.

(Supplementary Table 10)

**Comparison of different Cox proportional hazards models and improved predictive performance due to addition of age discrepancy to models**

| Model | Covariates in Cox model | AIC | Compared with | Performance improvement | | | | | | | | | | | |
|---|---|---|---|---|---|---|---|---|---|---|---|---|---|---|---|
| | | | | IDI | | | | cNRI | | | | MI | | | |
| | | | | Estimate | lower 95% | upper 95% | P value | Estimate | lower 95% | upper 95% | P value | Estimate | lower 95% | upper 95% | P value |
| Model 1 | Age + Sex + BMI + LVEF +log(NT-proBNP) + Hb + eGFR | 7608.6 | | | | | | | | | | | | | |
| Model 2 | Age + Sex + BMI + LVEF +log(NT-proBNP) + Hb + eGFR + Age_discrepancy | 7206.6 | vs Model 1 | 0.0104 | 0.0007 | 0.0245 | 0.020 | 0.134 | 0.025 | 0.201 | 0.010 | 0.0110 | 0.0008 | 0.0250 | 0.020 |
| Model 3 | Age + Sex + BMI + LVEF +log(NT-proBNP) + Hb + eGFR + CXR_Abnormality | 7204.4 | | | | | | | | | | | | | |

Model 4	Age + Sex + BMI + LVEF +log(NT-proBNP) + Hb + eGFR + CXR_Abnormality + Age_discrepancy	7198.9	vs Model 3	0.0105	0.0007	0.0251	0.030	0.135	0.034	0.202	0.030	0.0110	0.0008	0.0260	0.030
Model 5	Age + Sex + BMI + LVEF +log(NT-proBNP) + Hb + eGFR + CXR_Abnormality_score	7206.1													
Model 6	Age + Sex + BMI + LVEF +log(NT-proBNP) + Hb + eGFR + CXR_Abnormality_score + Age_discrepancy	7200.4	vs Model 5	0.0105	0.0008	0.0250	0.020	0.121	0.036	0.197	0.020	0.0120	0.0008	0.0250	0.020

REVIEWERS' COMMENTS:

Reviewer #3 (Remarks to the Author):

As before, I would like to thank the authors for their thorough responses to all of the reviewer comments. I feel that this round of revisions has even further strengthened the manuscript, and I do not have any further substantial comments for revision. I only have one minor comment regarding the wording of the response to Reviewer 2, Major Comment 1:

"Examples of heatmap images are shown in Fig. 3 and Supplementary Fig. 6. The model mainly focused on the top of the mediastinum and periphery of the rib cage regardless of the pathology present in the CXRs, where the shape and calcification of the aorta seemed to affect the estimation."

Consider rephrasing to something similar to "The model mainly focused on the top of the mediastinum as well as bony features in the sternum, clavicles, and shoulders regardless of the pathology present in the CXRs. This is hypothesis provoking that perhaps there are features in these areas (e.g. joint spaces or shape and calcification of the aorta) that are important for predicting CXR age." This is a softer statement that avoids attributing the predictions definitely to anatomical features, as the saliency mapping only tells us what regions of the image were important for prediction but we can't make assumptions about what anatomical features within these regions were important.

REVIEWER COMMENTS

Reviewer #3 (Remarks to the Author):

As before, I would like to thank the authors for their thorough responses to all of the reviewer comments. I feel that this round of revisions has even further strengthened the manuscript, and I do not have any further substantial comments for revision. I only have one minor comment regarding the wording of the response to Reviewer 2, Major Comment 1:

"Examples of heatmap images are shown in Fig. 3 and Supplementary Fig. 6. The model mainly focused on the top of the mediastinum and periphery of the rib cage regardless of the pathology present in the CXRs, where the shape and calcification of the aorta seemed to affect the estimation."

Consider rephrasing to something similar to "The model mainly focused on the top of the mediastinum as well as bony features in the sternum, clavicles, and shoulders regardless of the pathology present in the CXRs. This is hypothesis provoking that perhaps there are features in these areas (e.g. joint spaces or shape and calcification of the aorta) that are important for predicting CXR age." This is a softer statement that avoids attributing the predictions definitely to anatomical features, as the saliency mapping only tells us what regions of the image were important for prediction but we can't make assumptions about what anatomical features within these regions were important.

Response: We sincerely thank the reviewer's careful reading and assessment of our revised manuscript. We agree with the reviewer's comment. We revised our manuscript accordingly.

Lines 373-377

The model mainly focused on the top of the mediastinum as well as bony features in the sternum, clavicles, and shoulders regardless of the pathology present in the CXRs (**Fig. 3** and **Supplementary Fig. 6**). This is hypothesis-provoking that perhaps there are features in these areas (e.g., joint spaces, cartilage or shape and calcification of the aorta) that are important for predicting X-ray age. This pattern is similar to the previously reported CXR-based age prediction model using DNN ^{24 25}, ...